# MPF-Bench : A Large Scale Dataset for SciML of Multi-Phase-Flows: Droplet and Bubble Dynamics

## Abstract

Multiphase fluid dynamics, such as falling droplets and rising bubbles, are critical to many industrial applications. However, simulating these phenomena efficiently is challenging due to the complexity of instabilities, wave patterns, and bubble breakup. This paper investigates the potential of scientific machine learning (SciML) to model these dynamics using neural operators and foundation models. We apply sequence-to-sequence techniques on a comprehensive dataset generated from 11,000 simulations, comprising 1 million time snapshots, produced with a well-validated Lattice Boltzmann method (LBM) framework. The results demonstrate the ability of machine learning models to capture transient dynamics and intricate fluid interactions, paving the way for more accurate and computationally efficient SciML-based solvers for multiphase applications.

## 1 Introduction

Flow behavior in multiphase flow is crucial for many industrial and chemical applications. In drug delivery, two-phase flow can be used to create uniform drug-loaded microspheres or microcapsules. These microcapsules can provide controlled and sustained release of drugs, improving therapeutic outcomes (Hernot & Klibanov, 2008; Sattari et al., 2020). Two-phase flows are also essential for rapid diagnostics and biochemical applications in lab-on-a-chip technologies (Haeberle & Zengerle, 2007; Mark et al., 2010). Discrete phase bubbles in microchannels, generated via T-junctions (Thorsen et al. (2001)), co-flowing systems (Cramer et al., 2004), or flow-focusing techniques (Anna et al., 2003), have a high surface-to-volume ratio, enhancing reaction efficiency and sensitivity. The shearing forces of the continuous phase precisely control bubble size and formation, which is crucial for device performance. By thoroughly understanding gas-liquid or liquid-liquid interactions, engineers can optimize mixing conditions (Schwesinger et al., 1996; Stroock et al., 2002; Tice et al., 2003) to enhance reaction rates, improve product consistency, and reduce energy consumption.

Bubbles (*lighter fluid volumes moving in a denser fluid medium*) and droplets (*heavier fluid volumes moving in a lighter fluid medium*) play an integral role in applications such as drug delivery and lab-on-a-chip technologies. The dynamics of droplets and bubbles exhibit significant complexity, primarily due to phenomena such as breakup, deformation, and surface tension. Firstly, the breakup of droplets and bubbles is a highly nonlinear and complex process governed by factors such as viscosity ratio, density ratio, and surface tension. For example, for high inertia flows, the fast and irregular breakup results in smaller and widely-distributed droplets; at low Reynolds numbers, laminar flow leads to a more even breakup and larger droplets (Eggers & Villermaux, 2008). Secondly, droplets can be deformed by shear and pressure forces. Various studies have shown that the Capillary number (Vananroye et al., 2008; Liu et al., 2022), Atwood number (Fakhari & Rahimian, 2010; Singh, 2020), and Reynolds number (Vontas et al., 2020; Xu et al., 2020; Seksinsky & Marshall, 2021) all have a significant impact on droplet deformation.

To better understand multiphase phenomena (both droplets and bubbles), researchers often perform a canonical simulation/experiment called the bubble rising case (Bhaga & Weber, 1981b; Hua & Lou, 2007; Hysing et al., 2009; Amaya-Bower & Lee, 2010; Aland & Voigt, 2012; Yuan et al., 2017; Khanwale et al., 2023), where a bubble is placed in a higher density fluid so that the bubble moves up due to buoyancy. Conversely, using a droplet of higher density causes the droplet to fall down due to gravity (Yang et al., 2021; Jalaal & Mehravaran, 2012). This canonical study is essential since it

provides insights into bubble dynamics and shape evolution, which are critical factors for optimizing industrial processes and improving numerical models in fluid dynamics research. Nonetheless, capturing the bubble-rising or droplet-falling phenomenon is a multiscale problem with forces acting at different scales, ranging from microscale molecular interactions to macroscale fluid dynamics. Therefore, high-fidelity simulations are essential to accurately resolve these interactions, particularly at the thin interfaces where precise capturing of surface tension and interfacial dynamics is critical.

Scientific Machine Learning (SciML) represents a powerful approach for addressing multiphase flow problems. SciML leverages the inherent physics to develop models that can learn from complex data and produce reliable predictions (Karniadakis et al., 2021; Hassan et al., 2023; M Silva et al., 2024). A key ingredient to training and accessing SciML solvers is a comprehensive dataset, which MPF-Bench is an example of such a benchmark dataset. It includes wave patterns, bubble and droplet dynamics, and breakup.

There are several approaches to using machine learning to solve scientific problems, including Physics-Informed Neural Networks (PINNs) (Raissi et al., 2019) and neural operators (Li et al., 2021; Raonić et al., 2023; Lu et al., 2021). However, PINNs suffer from hard convergence and high generalization error (Rathore et al., 2024). In this paper, we focus on using neural operators and foundation models which use supervised learning. MPF-Bench has three major features:

- **Scientific machine learning evaluations**: We test our dataset on several neural operators and foundation models using the sequence-to-sequence time series concatenation technique. Our dataset serves as a good test for these models to evaluate their ability to learn multiscale physics data.
- **Extensive amount of data**: Our dataset includes 11,000 simulations in 2D and 3D with over 1 million time-series snapshots. This extensive volume of data allows for robust training of SciML models, which will help in advancing the development of accurate and reliable SciML models for multiphase flow dynamics.
- **Multiphase simulations**: We conduct simulations of rising bubbles and falling droplets, solving the Navier-Stokes equations coupled with the Allen-Cahn equation. This approach captures considerable physical phenomena, including breakup and deformation.

**Our Contributions:** We summarize our main contributions below:

- Six neural operators and foundation models trained on our data i.e., predicting concentration, velocity, and pressure solution fields using previous time solutions as input to the models. To our knowledge, no study has evaluated the performance of neural operators and foundation models on multiphase flows.
- Our dataset features 11,000 simulations and over 1 million time-series snapshots, with variations in density ratio, viscosity ratio, Reynolds number, and Bond number. This extensive dataset encompasses many phenomena, ranging from subtle surface deformations in bubble oscillations to full bubble breakups driven by surface tension and density ratio variations. The richness and breadth of this dataset offer deep insights into the intricate dynamics of multiphase flows, making it a valuable resource for advancing research in this field. We provide our dataset as a benchmark for others interested in developing and evaluating SciML models. Additional details can also be found in our website.

## 2 RELATED WORK

The Stanford Multiphase Flow Database (SMFD) used in (Chaari et al., 2018), the flow experiment dataset (Al-Dogail & Gajbhiye, 2021), and the BubbleML dataset (Hassan et al., 2023) are resources for understanding multiphase flow dynamics.

The SMFD features 5659 measurements across a range of gas and liquid properties, pipe characteristics, and operational conditions. This dataset, derived from laboratory and field sources, supports various pipe inclinations and flow patterns. SMFD covers different flow regimes, including stratified, slug, and annular flows. However, it does not appear publicly available, so we cannot identify the number of individual snapshots in this dataset.

The flow experiment dataset (Al-Dogail & Gajbhiye, 2021) focuses on the effects of density, viscosity, and surface tension on two-phase flow regimes and pressure drops in horizontal pipes. The 2904 measurements from air-liquid system experiments provide insights into fluid properties' influence on

**Table 1:** *Comparison of public Multiphase Flow Datasets*

| Name | Samples | Snapshots | Scope | Sources | Ranges of material properties |
|------|---------|-----------|-------|---------|-------------------------------|
| **Flow Experiment Dataset** | 2904 | 2904 | Horizontal pipes, effects of density, viscosity, surface tension | Controlled lab environment | $\rho$: [1, 1.5] gm/cc, $\mu$: [1, 3.1] cP, Surface tension = [32, 70] mN/m |
| **BubbleML** | 79 | 7641 | pool boiling, flow boiling, and sub-cooled boiling | 2D and 3D Numerical simulations based on Flash-X | Re = 0.0042, $\rho^* = 0.0083$, $\mu^* = 1$, Pr = 8.4, We = 1, Fr = [1, 100] |
| **MPF-Bench** | 11000 | > 1 million | Droplet and bubble dynamics | 2D and 3D Simulations using LBM | $\rho^* : [10, 1000]$, $\mu^* : [1, 100]$, Bo : $[10, 500]$, Re : $[10, 1000]$ |

flow regimes and pressure drops. This dataset's development of flow regimes and pressure contour maps enhances the understanding of fluid behavior in horizontal two-phase flows.

Additionally, the BubbleML dataset (Hassan et al., 2023) is a data collection focused on multiphysics phase change phenomena generated through physics-driven simulations, providing ground truth information for various boiling scenarios, including nucleate pool boiling, flow boiling, and sub-cooled boiling. We summarize these and other databases alongside our dataset in Table 1.

## 3 MULTI-PHASE FLOW (MPF) BENCH

We present the MPF-Bench dataset, encompassing 5500 bubble rise and 5500 droplet flow simulations, with each simulation containing 100 time-snapshots, making it, to our knowledge, two orders of magnitude larger – in terms of number of time-snapshots – than any existing multiphase flow dataset. This dataset features 2D and 3D transient simulations, capturing a spectrum of flow behaviors influenced by surface tension and density/viscosity ratios. MPF-Bench includes scenarios from bubble oscillations with minor surface deformations to complete bubble breakup, offering a comprehensive resource for studying bubble rise and droplet fall dynamics.

### 3.1 PROBLEM DEFINITION: INITIAL AND BOUNDARY CONDITIONS, AND OUTPUTS

We consider 2D and 3D simulations of bubble rise and droplet fall simulations using the lattice Boltzmann method. The domain sizes for 2D and 3D are $[256, 512]$ and $[128, 256, 128]$ lattice units, respectively. For 2D simulations, the bubble is initially centered at $(64, 64)$ and the droplet is centered at $(128, 384)$. In 3D, the bubble is centered at $(64, 64, 64)$ while the droplet is centered at $(128, 384, 64)$. The initial diameter $D_0$ for both problems is set to 128 lattice units in 2D and 64 lattice units in 3D. The boundary conditions are set to free-slip on the side walls and periodic at the top and bottom as illustrated in Figure 1. This problem is driven mainly by the density and viscosity ratio of the two phases in addition to the Reynolds and Bond numbers. The Reynolds number measures the ratio of inertial forces to viscous forces, while the Bond number measures the ratio of gravitational forces to surface tension forces. Below is the definition of these four dimensionless numbers:

$$\rho^* = \frac{\rho_h}{\rho_l}, \quad \mu^* = \frac{\mu_h}{\mu_l}, \quad \text{Re}_h = \frac{\sqrt{g_y \rho_h (\rho_h - \rho_l) D^3}}{\mu_h}, \quad \text{Bo} = \frac{g_y (\rho_h - \rho_l) D^2}{\sigma} \quad (1)$$

where $h$ and $l$ indices refer to the heavy and light fluids, respectively. We have selected random, dimensionless numbers uniformly to ensure the entire defined range is covered. The outputs of the simulations are the interface indicator $(c)$, velocity components $(u, v, w)$, pressure $(p)$, and density $(\rho)$, which provide insights into the dynamics of multiphase flow and the interactions between the phases.

We selected a few representative cases from our dataset to illustrate the key physics of droplet and bubble dynamics (see Table 2). As shown in Figure 2 and Figure 4, these cases highlight how variations in Bond number, Reynolds number, and density ratio affect droplet deformation and breakup patterns. Each case reveals distinct fluid behaviors, enhancing our understanding of the complex, nonlinear dynamics. The streamlines around the bubble and droplet, depicted in Figure 3 and Figure 5, further illustrate how these physical parameters influence droplet breakup and stability across 3D and 2D flows."

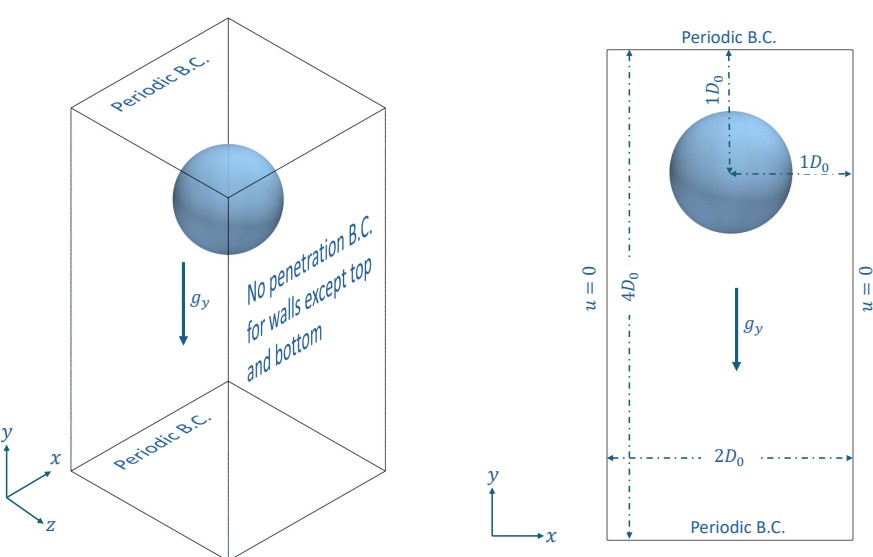

**Figure 1:** *Boundary conditions for the simulation of a falling droplet. The left panel illustrates the 3D case, while the right panel illustrates the 2D case.*

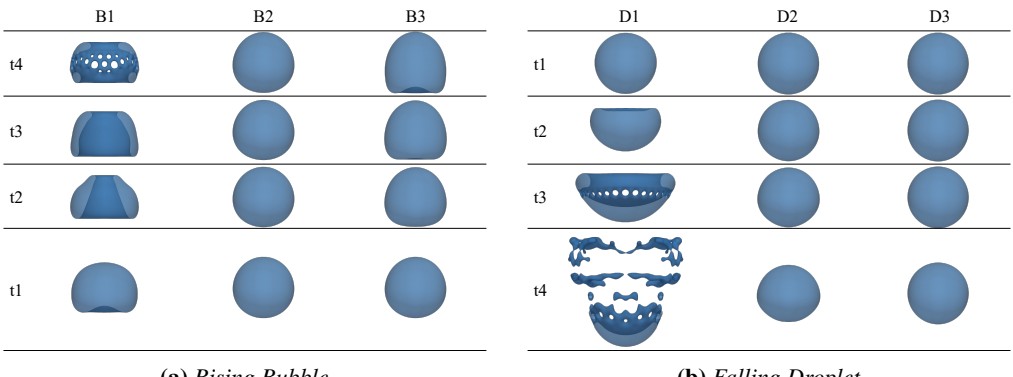

**(a)** *Rising Bubble*  **(b)** *Falling Droplet*

**Figure 2:** *(a) Snapshot of a 3D rising bubble and (b) snapshot of a 3D falling droplet. The properties of the fluids for each case are detailed in Table 2.*

## 3.2 SIMULATION FRAMEWORK AND COMPUTE EFFORT

Our simulation framework employs a highly parallel, in-house Lattice Boltzmann code, utilizing one of the most accurate two-phase models, the phase field model, to capture the complexities of the interface. The code has been rigorously tested across various problems, with validation results provided in Section A.3. For 2D simulations, we used a uniform lattice grid of $256 \times 512$, while for 3D simulations, the domain was set to $128 \times 256 \times 128$. We achieved high parallelization by distributing the computation across 12 Nvidia A100-SXM4 80GB GPUs. The total computational cost for 2D and 3D cases was approximately 4,000 GPU hours. We use the ParaView tool (Ayachit, 2015) to visualize and understand our dataset.

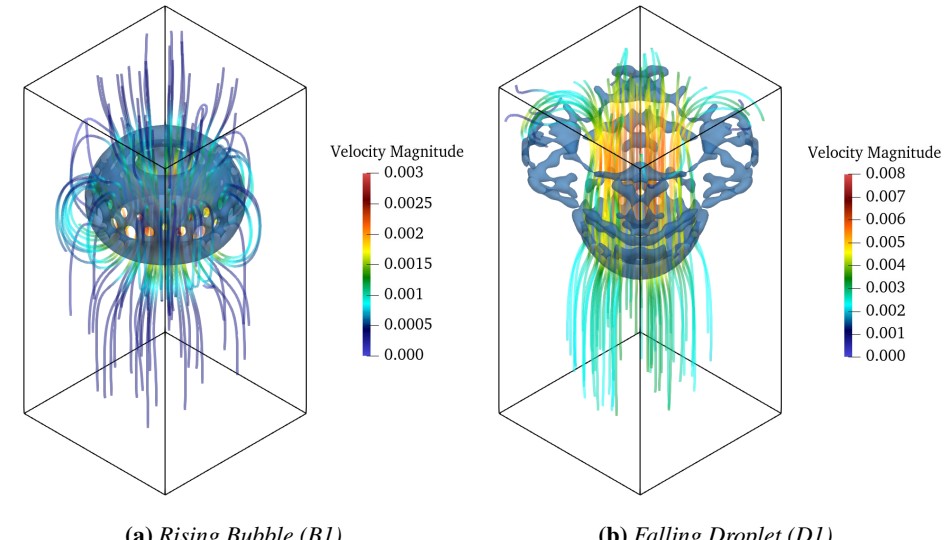

**(a)** *Rising Bubble (B1)*       **(b)** *Falling Droplet (D1)*

**Figure 3:** *Streamlines of a 3D rising bubble (a) and a 3D falling droplet (b), with colors indicating the magnitude of velocity. The properties of the fluids for each case are detailed in Table 2.*

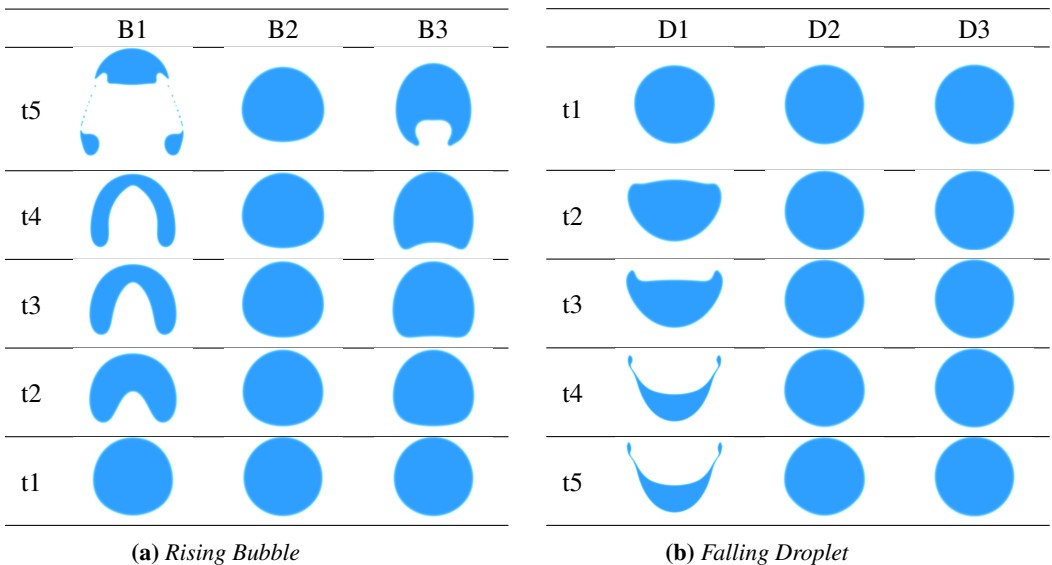

**(a)** *Rising Bubble*       **(b)** *Falling Droplet*

**Figure 4:** *(a) Snapshot of a 2D rising bubble and (b) snapshot of a 2D falling droplet. The properties of the fluids for each case are detailed in Table 2.*

### 3.3 METADATA

**Input Fields**: We have provisioned the following dimensionless quantities as inputs to our study as defined in Section 3.1. These are the Density Ratio ($\rho^*$), Viscosity Ratio ($\mu^*$), Bond Number ($Bo$), and Reynolds Number ($Re$). Since these are scalar values, we feed them to the neural network by creating a constant field with a dimension consistent with the number of samples, in this case, 10,000 in 2D and 1000 in 3D.

**Output Fields**: In analyzing multiphase flow problems, we are interested in solving the governing PDEs to obtain solutions at every point in the domain's interior for certain cardinal fields. For a 2D solution domain, these are: $c$ - interface indicator, $u$ - velocity in $x$ direction, $v$ - velocity in $y$ direction, $p$ - pressure. Additionally, because this is a time-dependent problem, we have these cardinal fields or a sequence of these fields distributed uniformly over time (100 time steps).

**Table 2:** *Material properties and nondimensional numbers of three bubble rise simulations (B1, B2, B3) and three droplet fall simulations (D1, D2, D3). The table shows the density ratio, viscosity ratio, Reynolds number, and Bond number of all six simulations.*

| case | B1 | B2 | B3 | D1 | D2 | D3 |
|---|---|---|---|---|---|---|
| Density Ratio ($\rho^*$) | $10^3$ | $10^3$ | $10^3$ | 10 | $10^3$ | $10^3$ |
| Viscosity Ratio ($\mu^*$) | $10^2$ | $10^2$ | $10^2$ | 1 | $10^2$ | $10^2$ |
| $Re$ | $5 \times 10^2$ | 10 | 10 | $10^3$ | $10^3$ | 10 |
| $Bo$ | $5 \times 10^2$ | 10 | $5 \times 10^2$ | $5 \times 10^2$ | $5 \times 10^2$ | 10 |

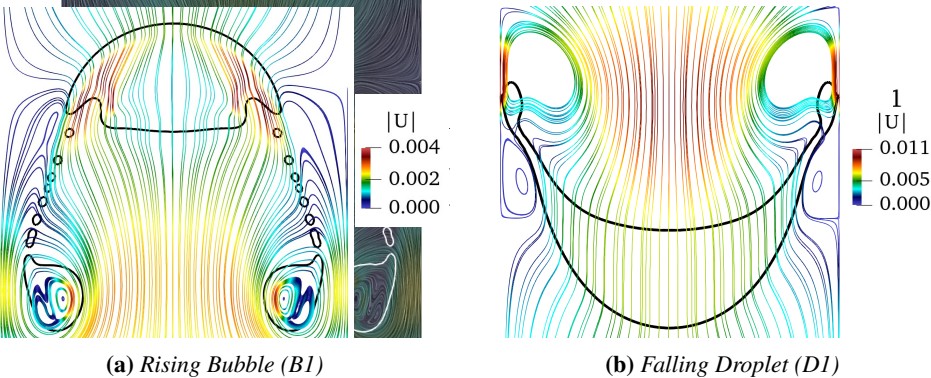

**(a)** *Rising Bubble (B1)*        **(b)** *Falling Droplet (D1)*

**Figure 5:** *Streamlines of a 2D rising bubble (a) and a 2D falling droplet (b), with colors indicating the magnitude of velocity.*

**Resolution**: We maintained the original resolution of our datasets, matching the Lattice Boltzmann simulation domain. This ensures the complete physics is presented to the Neural Operator and allows direct comparison with Lattice Boltzmann method simulations. Our datasets are published at $256 \times 512$ resolution for 2D and $128 \times 256 \times 128$ for 3D simulations.

**Dataset Format**: For both the bubble and droplet datasets, we have released a single file for each sample. This decision was taken with the view to allow for maximum flexibility to the end user in deciding what and how many time steps they want to use to train their models, as these time-dependent problems often take the shape of sequence to sequence formulations. In 2D, the resulting `.npz` files take the form:

$$[\textbf{number\_of\_time\_steps}][\textbf{number\_of\_channels}][\textbf{resolution\_y}][\textbf{resolution\_x}]$$

whereas in 3D, [**resolution_z**] incorporated as an additional dimension for depth. In this study, we have released a total of 11,000 samples spread across two families of datasets. Table 3 provides a detailed formulaic description of the packaging of the input and output `numpy` tensors for both these families:

**Level of Difficulty**: We provide Table 4 to help users select datasets based on varying difficulty levels. The dataset includes key parameters like Reynolds number (Re), Density Ratio, Viscosity Ratio, and Bond Number (Bo), with a difficulty classification to guide users. This classification reflects the complexity of interface deformations, making it easier to choose suitable cases for model training and evaluation.

## 3.4 EVALUATION METRICS AND TEST DATASET ANALYSIS

We assess the performance of the trained neural operators and foundation models using two primary metrics: Mean Squared Error (MSE) and relative $L_2$ error. Our models are trained on a random selection of 1000 samples from the bubble dataset. To manage the transient nature of the data, we employ sequence-to-sequence and sequence-to-field mappings, where the solution fields at various time steps are concatenated and fed sequentially as input into the models. We skip every 4 timesteps to have more dynamics in the dataset bringing down the total number of timesteps to 25. The models are evaluated on six distinct test subsets ($S1$ through $S6$) as outlined below:

**Table 3:** *Formulaic description of the input and output tensors. 5000 - sample size for the dataset. 101 - number of time steps in the simulation. $x, y$ - The $x, y$ dimension of a field. E.g., $Y[0, 100, 1, :, :]$ indicates the pointwise $v$ velocity over the entire grid of size $256 \times 512$ for the first sample at time step 100.*

| Dataset | Dim. | Input Tensor | Output Tensor |
|---------|------|--------------|---------------|
| Droplet | 2 | $X[5000][\rho^*, \mu^*, Bo, Re]$ | $Y[5000][101][c, u, v, p, \rho][y][x]$ |
| Bubble | 2 | $X[5000][\rho^*, \mu^*, Bo, Re]$ | $Y[5000][101][c, u, v, p, \rho][y][x]$ |
| Droplet | 3 | $X[500][\rho^*, \mu^*, Bo, Re]$ | $Y[500][51][c, u, v, w, p, \rho][z][y][x]$ |
| Bubble | 3 | $X[500][\rho^*, \mu^*, Bo, Re]$ | $Y[500][51][c, u, v, w, p, \rho][z][y][x]$ |

**Table 4:** *Dataset parameters with difficulty levels for selecting appropriate cases based on Reynolds number, Density Ratio, Viscosity Ratio, and Bond Number.*

| Density Ratio | Viscosity Ratio | $Bo$ Number | $Re$ Number | Difficulty |
|---------------|-----------------|-------------|-------------|------------|
| High | High | High | High | Challenging |
| High | High | Low | Low | Easy |
| High | High | Low | Low | Moderate |
| High | High | Low | High | Moderate |
| High | High | High | High | Challenging |
| Low | Low | High | High | Challenging |
| Low | Low | Low | Low | Easy |
| Low | Low | High | High | Easy |
| Low | Low | Low | High | Moderate |

- **Sequence-to-field:** We set up 3 different inputs for subsets $S1$, $S2$, and $S3$. We input the solution at timestep $t1$, sequences $t1$ to $t3$, and sequences $t1$ to $t5$. The output solution for $S1$, $S2$, and $S3$ is the corresponding next time snapshot for each subset.
- **Sequence-to-sequence:** In this case, the output is not a single time snapshot but a sequence of solutions. We input the solution over sequences $t1$ to $t3$, $t1$ to $t5$, and $t1$ to $t8$ respectively. The output for $S4$, $S5$, and $S6$ is a sequence of three time snapshots corresponding to the next solutions of each subset respectively.

## 4 EXPERIMENTS

Neural Operators represent a novel class of deep learning architectures specifically designed to learn functional solutions to partial differential equations (PDEs). Unlike traditional methods that focus on finding a specific solution for a fixed set of parameters, Neural Operators are capable of learning generalized solutions to PDEs. While these frameworks have demonstrated notable success in modeling single-phase fluid flow, there is limited research on their accuracy in capturing multi-phase flows. Multi-phase flows present additional challenges due to phenomena like bubble or droplet breakup, coalescence, and shape oscillations. In this context, we aim to evaluate the performance of several Neural Operators and foundation models in learning these intricate fluid dynamics.

We report baseline results for training a suite of the most common neural PDE solvers. We studied the following Neural Operators and Foundation Models, reporting results on the 2D bubble case: (a) Fourier Neural Operator (FNO) (Li et al., 2021), (b) Convolutional Neural Operators (CNO) (Raonić et al., 2023), (c) DeepONet (Lu et al., 2021), (d) UNet (Ronneberger et al., 2015), (e) scOT (Herde et al., 2024), (f) Poseidon (Herde et al., 2024). For training, we adhered closely to the published code examples. All the aforementioned models were trained on a single A100 80GB GPU using the Adam optimizer with a learning rate of $10^{-3}$ and were run for 200 epochs. The validation loss for all models converged and stabilized by 200 epochs.

Table 5 and Table 6 compare the Mean Squared Error (MSE) and relative $L_2$ error for sequence-to-field and sequence-to-sequence predictions across various models on the six bubble rise datasets (S1-S6). These results highlight the performance differences between models in predicting the solution fields for different data subsets (S1-S6). Notably, CNO generally outperforms the other models in predicting the concentration field, demonstrating a clear advantage. Additionally, the model's ability to capture the solution fields improves as more time snapshots are incorporated into

**Table 5:** *Comparison of mean squared error MSE and relative $L_2$ Error for Sequence-to-Field predictions using UNet, DeepONet, FNO, CNO, scOT, and Poseidon on Bubble Datasets (S1-S3).*

| Model | Channel | S1 | | S2 | | S3 | |
|---|---|---|---|---|---|---|---|
| | | MSE | $L_2$ | MSE | $L_2$ | MSE | $L_2$ |
| UNet | $c$ | $\mathbf{2.60 \times 10^{-2}}$ | $\mathbf{2.59 \times 10^{-2}}$ | $8.40 \times 10^{-3}$ | $8.07 \times 10^{-3}$ | $9.56 \times 10^{-3}$ | $9.04 \times 10^{-3}$ |
| | $u$ | $8.80 \times 10^{-5}$ | $3.13 \times 10^{0}$ | $8.00 \times 10^{-6}$ | $2.04 \times 10^{0}$ | $1.00 \times 10^{-5}$ | $2.09 \times 10^{0}$ |
| | $v$ | $1.00 \times 10^{-6}$ | $1.56 \times 10^{0}$ | $1.00 \times 10^{-6}$ | $7.84 \times 10^{-1}$ | $1.00 \times 10^{-6}$ | $8.76 \times 10^{-1}$ |
| | $p$ | $1.00 \times 10^{-6}$ | $2.74 \times 10^{2}$ | $1.00 \times 10^{-6}$ | $3.23 \times 10^{2}$ | $1.00 \times 10^{-6}$ | $5.17 \times 10^{2}$ |
| DeepONet | $c$ | $2.66 \times 10^{-2}$ | $2.65 \times 10^{-2}$ | $1.01 \times 10^{-1}$ | $1.01 \times 10^{-1}$ | $1.18 \times 10^{-1}$ | $1.18 \times 10^{-1}$ |
| | $u$ | $9.10 \times 10^{-5}$ | $6.71 \times 10^{0}$ | $1.27 \times 10^{-3}$ | $1.24 \times 10^{0}$ | $1.71 \times 10^{-3}$ | $9.18 \times 10^{-1}$ |
| | $v$ | $1.00 \times 10^{-6}$ | $8.68 \times 10^{-1}$ | $1.00 \times 10^{-6}$ | $5.35 \times 10^{-1}$ | $1.00 \times 10^{-6}$ | $8.89 \times 10^{-1}$ |
| | $p$ | $1.00 \times 10^{-6}$ | $1.89 \times 10^{2}$ | $1.00 \times 10^{-6}$ | $2.43 \times 10^{2}$ | $1.00 \times 10^{-6}$ | $1.66 \times 10^{2}$ |
| FNO | $c$ | $2.72 \times 10^{-2}$ | $2.68 \times 10^{-2}$ | $9.73 \times 10^{-3}$ | $8.97 \times 10^{-3}$ | $2.10 \times 10^{-2}$ | $1.98 \times 10^{-2}$ |
| | $u$ | $9.30 \times 10^{-5}$ | $8.56 \times 10^{0}$ | $1.00 \times 10^{-5}$ | $2.77 \times 10^{0}$ | $4.80 \times 10^{-5}$ | $5.29 \times 10^{-1}$ |
| | $v$ | $1.00 \times 10^{-6}$ | $3.43 \times 10^{0}$ | $1.00 \times 10^{-6}$ | $1.09 \times 10^{0}$ | $2.00 \times 10^{-6}$ | $2.02 \times 10^{0}$ |
| | $p$ | $1.00 \times 10^{-6}$ | $7.18 \times 10^{2}$ | $1.00 \times 10^{-6}$ | $7.44 \times 10^{2}$ | $2.00 \times 10^{-6}$ | $1.04 \times 10^{3}$ |
| CNO | $c$ | $2.62 \times 10^{-2}$ | $2.60 \times 10^{-1}$ | $\mathbf{5.89 \times 10^{-3}}$ | $\mathbf{5.65 \times 10^{-3}}$ | $\mathbf{9.41 \times 10^{-3}}$ | $\mathbf{9.00 \times 10^{-3}}$ |
| | $u$ | $8.80 \times 10^{-5}$ | $5.04 \times 10^{0}$ | $4.00 \times 10^{-6}$ | $1.60 \times 10^{0}$ | $1.00 \times 10^{-5}$ | $1.79 \times 10^{0}$ |
| | $v$ | $1.00 \times 10^{-6}$ | $1.73 \times 10^{0}$ | $1.00 \times 10^{-6}$ | $5.07 \times 10^{-1}$ | $1.00 \times 10^{-6}$ | $9.19 \times 10^{-1}$ |
| | $p$ | $1.00 \times 10^{-6}$ | $4.46 \times 10^{2}$ | $1.00 \times 10^{-6}$ | $2.24 \times 10^{2}$ | $1.00 \times 10^{-6}$ | $3.96 \times 10^{2}$ |
| scOT | $c$ | $2.76 \times 10^{-2}$ | $2.68 \times 10^{-2}$ | $1.77 \times 10^{-2}$ | $1.68 \times 10^{-2}$ | $2.23 \times 10^{-2}$ | $2.17 \times 10^{-2}$ |
| | $u$ | $1.29 \times 10^{1}$ | $1.29 \times 10^{1}$ | $3.50 \times 10^{-5}$ | $3.91 \times 10^{0}$ | $5.80 \times 10^{-5}$ | $2.82 \times 10^{0}$ |
| | $v$ | $5.24 \times 10^{0}$ | $5.24 \times 10^{0}$ | $1.00 \times 10^{-6}$ | $2.31 \times 10^{0}$ | $1.00 \times 10^{-6}$ | $1.75 \times 10^{0}$ |
| | $p$ | $9.65 \times 10^{2}$ | $9.65 \times 10^{2}$ | $2.00 \times 10^{-6}$ | $8.12 \times 10^{2}$ | $2.00 \times 10^{-6}$ | $7.80 \times 10^{2}$ |
| Poseidon | $c$ | $2.87 \times 10^{-2}$ | $2.79 \times 10^{-1}$ | $3.34 \times 10^{-2}$ | $3.01 \times 10^{-2}$ | $2.49 \times 10^{-2}$ | $2.23 \times 10^{-2}$ |
| | $u$ | $1.00 \times 10^{-4}$ | $1.16 \times 10^{1}$ | $1.14 \times 10^{-4}$ | $1.31 \times 10^{1}$ | $6.10 \times 10^{-5}$ | $8.28 \times 10^{0}$ |
| | $v$ | $2.00 \times 10^{-6}$ | $5.86 \times 10^{0}$ | $1.10 \times 10^{-5}$ | $4.05 \times 10^{0}$ | $5.00 \times 10^{-6}$ | $2.83 \times 10^{0}$ |
| | $p$ | $2.00 \times 10^{-6}$ | $1.04 \times 10^{3}$ | $6.00 \times 10^{-6}$ | $2.35 \times 10^{3}$ | $4.00 \times 10^{-6}$ | $1.94 \times 10^{3}$ |

**Table 6:** *Comparison of mean squared error MSE and relative $L_2$ Error for Sequence-to-Sequence predictions using UNet, DeepONet, FNO, CNO, scOT, and Poseidon on Bubble Datasets (S4-S6).*

| Model | Channel | S4 | | S5 | | S6 | |
|---|---|---|---|---|---|---|---|
| | | MSE | $L_2$ | MSE | $L_2$ | MSE | $L_2$ |
| UNet | $c$ | $3.27 \times 10^{-2}$ | $2.87 \times 10^{-2}$ | $3.93 \times 10^{-2}$ | $3.31 \times 10^{-2}$ | $7.34 \times 10^{-2}$ | $6.58 \times 10^{-2}$ |
| | $u$ | $1.35 \times 10^{-4}$ | $3.54 \times 10^{0}$ | $2.02 \times 10^{-4}$ | $3.10 \times 10^{0}$ | $6.65 \times 10^{-4}$ | $3.90 \times 10^{0}$ |
| | $v$ | $1.00 \times 10^{-6}$ | $1.38 \times 10^{0}$ | $1.00 \times 10^{-6}$ | $1.45 \times 10^{0}$ | $1.00 \times 10^{-6}$ | $9.54 \times 10^{-1}$ |
| | $p$ | $1.00 \times 10^{-6}$ | $6.76 \times 10^{2}$ | $1.00 \times 10^{-6}$ | $6.30 \times 10^{2}$ | $1.00 \times 10^{-6}$ | $7.79 \times 10^{2}$ |
| DeepONet | $c$ | $1.64 \times 10^{-1}$ | $1.60 \times 10^{-1}$ | $2.03 \times 10^{-1}$ | $1.99 \times 10^{-1}$ | $1.94 \times 10^{-1}$ | $1.92 \times 10^{-1}$ |
| | $u$ | $3.33 \times 10^{-3}$ | $3.15 \times 10^{0}$ | $5.04 \times 10^{-3}$ | $1.64 \times 10^{0}$ | $4.57 \times 10^{-3}$ | $1.32 \times 10^{0}$ |
| | $v$ | $1.00 \times 10^{-6}$ | $1.04 \times 10^{0}$ | $1.00 \times 10^{-6}$ | $7.59 \times 10^{-1}$ | $1.00 \times 10^{-6}$ | $3.83 \times 10^{-1}$ |
| | $p$ | $1.00 \times 10^{-6}$ | $7.30 \times 10^{2}$ | $1.00 \times 10^{-6}$ | $3.53 \times 10^{2}$ | $1.00 \times 10^{-6}$ | $3.17 \times 10^{2}$ |
| FNO | $c$ | $\mathbf{1.16 \times 10^{-2}}$ | $\mathbf{1.10 \times 10^{-2}}$ | $2.33 \times 10^{-2}$ | $2.20 \times 10^{-2}$ | $4.24 \times 10^{-2}$ | $4.00 \times 10^{-2}$ |
| | $u$ | $1.70 \times 10^{-5}$ | $1.00 \times 10^{0}$ | $6.50 \times 10^{-5}$ | $2.98 \times 10^{0}$ | $2.28 \times 10^{-4}$ | $1.13 \times 10^{0}$ |
| | $v$ | $1.00 \times 10^{-6}$ | $4.50 \times 10^{-1}$ | $1.00 \times 10^{-6}$ | $1.28 \times 10^{0}$ | $1.00 \times 10^{-6}$ | $3.98 \times 10^{-1}$ |
| | $p$ | $1.00 \times 10^{-6}$ | $1.38 \times 10^{2}$ | $1.00 \times 10^{-6}$ | $7.87 \times 10^{2}$ | $1.00 \times 10^{-6}$ | $2.69 \times 10^{2}$ |
| CNO | $c$ | $1.74 \times 10^{-2}$ | $1.69 \times 10^{-2}$ | $\mathbf{1.72 \times 10^{-2}}$ | $\mathbf{1.63 \times 10^{-2}}$ | $\mathbf{3.78 \times 10^{-2}}$ | $\mathbf{3.53 \times 10^{-2}}$ |
| | $u$ | $3.70 \times 10^{-5}$ | $1.51 \times 10^{0}$ | $3.60 \times 10^{-5}$ | $1.37 \times 10^{0}$ | $1.76 \times 10^{-4}$ | $1.74 \times 10^{0}$ |
| | $v$ | $1.00 \times 10^{-6}$ | $6.87 \times 10^{-1}$ | $1.00 \times 10^{-6}$ | $6.85 \times 10^{-1}$ | $1.00 \times 10^{-6}$ | $5.98 \times 10^{-1}$ |
| | $p$ | $1.00 \times 10^{-6}$ | $2.84 \times 10^{2}$ | $1.00 \times 10^{-6}$ | $2.93 \times 10^{2}$ | $1.00 \times 10^{-6}$ | $4.22 \times 10^{2}$ |
| scOT | $c$ | $3.85 \times 10^{-2}$ | $3.69 \times 10^{-2}$ | $3.92 \times 10^{-2}$ | $3.79 \times 10^{-2}$ | $6.27 \times 10^{-2}$ | $6.09 \times 10^{-2}$ |
| | $u$ | $1.73 \times 10^{-4}$ | $6.73 \times 10^{0}$ | $1.80 \times 10^{-4}$ | $5.73 \times 10^{0}$ | $4.85 \times 10^{-4}$ | $5.31 \times 10^{0}$ |
| | $v$ | $3.00 \times 10^{-6}$ | $2.78 \times 10^{0}$ | $3.00 \times 10^{-6}$ | $2.12 \times 10^{0}$ | $2.00 \times 10^{-6}$ | $1.93 \times 10^{0}$ |
| | $p$ | $3.00 \times 10^{-6}$ | $1.48 \times 10^{3}$ | $3.00 \times 10^{-6}$ | $1.34 \times 10^{3}$ | $4.00 \times 10^{-6}$ | $1.21 \times 10^{3}$ |
| Poseidon | $c$ | $3.06 \times 10^{-2}$ | $2.84 \times 10^{-2}$ | $3.33 \times 10^{-2}$ | $3.17 \times 10^{-2}$ | $5.99 \times 10^{-2}$ | $5.79 \times 10^{-1}$ |
| | $u$ | $1.01 \times 10^{-4}$ | $8.26 \times 10^{0}$ | $1.26 \times 10^{-4}$ | $6.49 \times 10^{0}$ | $4.30 \times 10^{-4}$ | $7.79 \times 10^{0}$ |
| | $v$ | $5.00 \times 10^{-6}$ | $3.51 \times 10^{0}$ | $4.00 \times 10^{-6}$ | $2.47 \times 10^{0}$ | $5.00 \times 10^{-6}$ | $2.16 \times 10^{0}$ |
| | $p$ | $5.00 \times 10^{-6}$ | $1.71 \times 10^{3}$ | $4.00 \times 10^{-6}$ | $1.48 \times 10^{3}$ | $5.00 \times 10^{-6}$ | $1.81 \times 10^{3}$ |

the model, highlighting the benefits of utilizing more temporal data in these predictions. Another interesting observation is the fact that vision transformer Scot is marginally outperforming the pre-trained version of Poseidon. This suggests that Poseidon being trained on single-phase phenomena makes learning multiphase flow harder and less accurate.

Furthermore, Figure 6 and Figure 7 illustrate field predictions of the concentration field $C$ using UNet, CNO, DeepONet, and Poseidon for sequence-to-field and sequence-to-sequence scenarios, respectively. These figures show that DeepONet performs poorly in both the sequence-to-field and sequence-to-sequence scenarios. Also, UNet's accuracy declines as the prediction horizon extends to longer time sequences, as shown in Figure 7. This may be attributed to UNet's architecture, which,

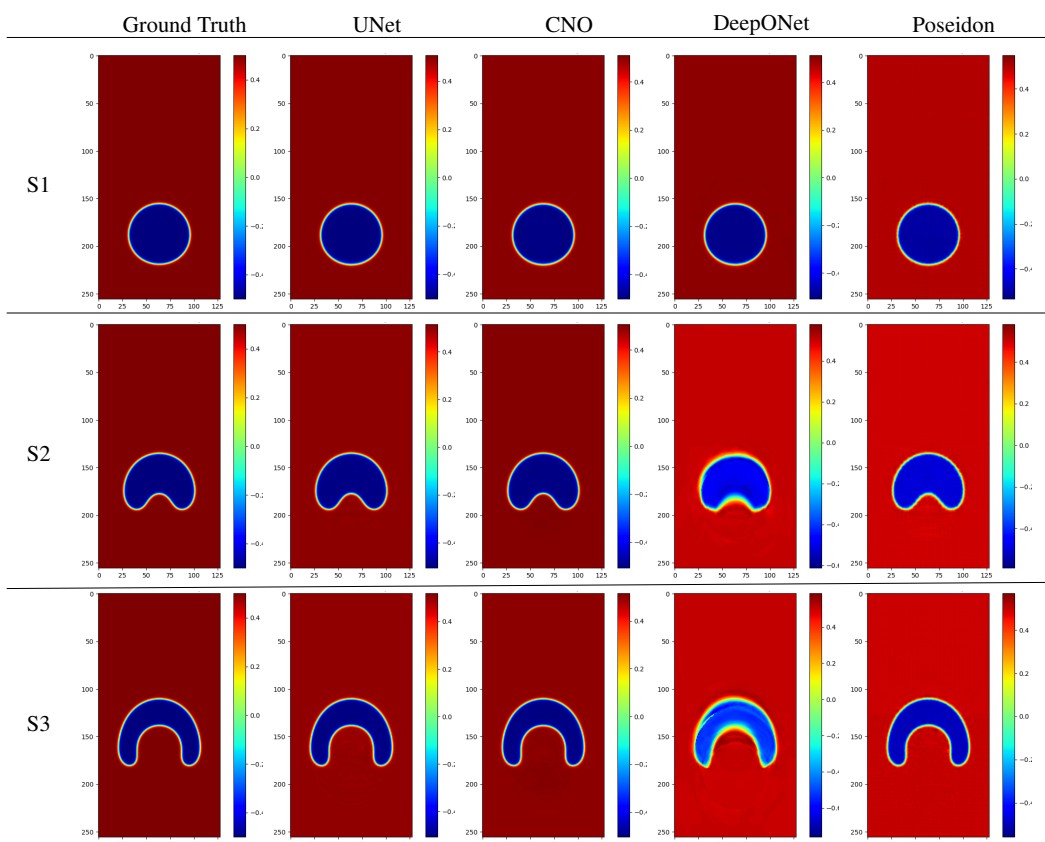

**Figure 6:** *The figure presents a comparison of sequence-to-field predictions for the concentration field C against the ground truth. The predictions are generated by four models: UNet, Convolutional Neural Operator (CNO), DeepONet, and Poseidon, across three data subsets (S1, S2, and S3). Each row corresponds to a different subset (S1, S2, or S3), while each column displays the predictions made by the respective models.*

unlike neural operators, is more adept at capturing local rather than global interface patterns. In contrast, CNO consistently delivers the best performance in both sequence-to-field and sequence-to-sequence predictions, reinforcing its capability in handling complex fluid dynamics over time. Also, Figure 7 shows that CNO can capture small bubble formation after breakup more accurately than other models.

## 5 CONCLUSIONS

In summary, we have introduced a comprehensive time series dataset comprising 10,000 simulations in 2D and 1,000 simulations in 3D, focusing on bubble rise and droplet fall dynamics. This dataset captures a wide range of two-phase flow phenomena, including simulations with density ratios as high as 1,000, Reynolds numbers up to 1,000, and Bond numbers up to 500. Using a subsample of 1,000 samples from the bubble dataset, we successfully trained neural operators and foundation models, demonstrating encouraging results. By feeding in more time snapshots to models, they can more accurately predict the trajectory of bubble dynamics. Specifically, we found that CNO outperformed other models in capturing fine-scale interfacial details. We also concluded that the foundation model Poseidon pre-trained on single-phase phenomena might not be effective in learning multiphase flow, which demonstrates the need to train foundation models on multiphase flow data.

**Limitations**: The dataset has the following constraints:

- **Different orders of magnitude for solution fields**: The dataset includes solution fields that span different orders of magnitude. This is evident in the large disparity between the mean squared error (MSE) and relative $L_2$ errors for different solution fields.

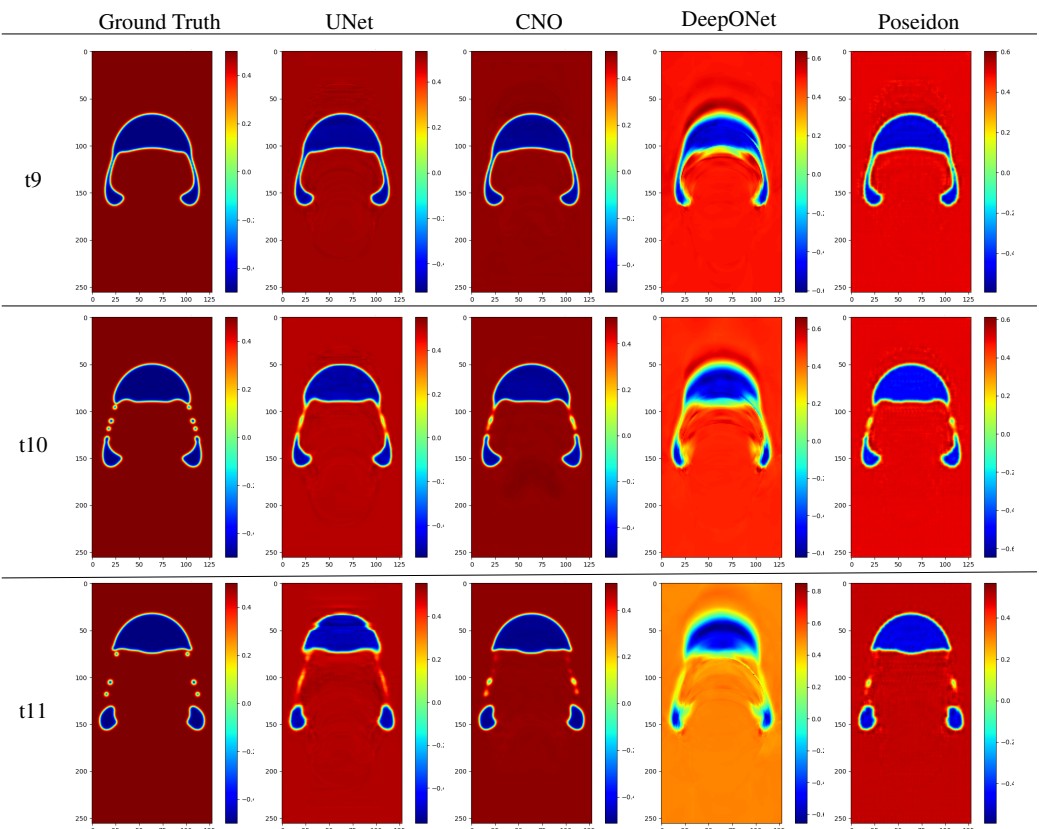

**Figure 7:** *The figure presents sequence-to-sequence predictions for the concentration field $C$ compared to the ground truth. Predictions are made by four models: UNet, Convolutional Neural Operator (CNO), DeepONet, and Poseidon. Each row represents different time steps (t9, t10, and t11) from dataset S6, while each column shows the predictions from the respective models.*

- **Limited 3D Simulations**: Due to the substantial computational cost, only a small number of 3D simulations were conducted, resulting in a more restricted set of 3D cases in the dataset.
- **Model fitting with a limited number of time steps**: GPU memory limitations constrained the number of time steps that could be fitted on a single GPU. As a result, we had to use a limited number of time snapshots. An alternative approach could involve using an auto-regressive model to model the time series for each sample.

## REPRODUCIBILITY STATEMENT

In this work, we introduce a dataset and provide detailed explanations of the methodology and mathematical framework used for data generation in the Appendix Section A. To evaluate the dataset, we applied various neural operators and foundation models, and the code for these models is available in our GitHub repository. The repository includes detailed instructions for easy reproducibility of our results. All experiments were conducted on Nvidia A100-SXM4 80GB. Please refer to the repository's README.md for complete instructions on replicating the model evaluations.

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

APPENDIX

## A  DETAILS OF THE CFD SIMULATION FRAMEWORK

Our computational framework employs the CUDA platform to implement the algorithms necessary for the Lattice Boltzmann Method (LBM). We achieve significant computational performance enhancements by leveraging CUDA's parallel processing capabilities. The primary performance bottleneck in GPU architectures is often the data transfer between GPU memory and unified CPU memory. To mitigate this, we minimize such data transfers, conducting them only when necessary for convergence checks or final output retrieval.

We utilize a single one-dimensional array in conjunction with macro functions to handle the substantial data volumes intrinsic to LBM simulations. This method optimizes memory usage and computational efficiency on the GPU, ensuring that we fully exploit the GPU's computational power and memory bandwidth. This strategy allows for the high-performance execution of LBM algorithms, crucial for large-scale simulations and complex fluid dynamics problems.

### A.1  FORMULATION OF NAVIER STOKES AND ALLEN CAHN EQUATIONS

Several lattice Boltzmann models, such as the Cahn-Hilliard and Allen-Cahn models, utilize interface tracking equations and are thus categorized as phase-field models (Penrose & Fife, 1990; Jacqmin, 1999). These models describe multiphase flows using a diffuse interface, with the Allen-Cahn equation commonly employed for this purpose (Allen & Cahn, 1976). In some studies, this approach is called the conservative phase-field LB model (Fakhari et al., 2019). The phase-field variable, $\phi$, which tracks the interface, ranges from 0 to 1, leading to the following expression for the phase-field equation (Chiu & Lin, 2011):

$$\frac{\partial \phi}{\partial t} + \nabla . (\phi u) = \nabla . \left[ M(\nabla \phi - \frac{1 - 4(\phi - \phi_0)^2}{\xi} \hat{n}) \right], \tag{2}$$

where $t$ represents time, $u$ is the velocity, $M$ denotes a positive constant for the mobility parameter, $\xi$ is the interfacial thickness, and $\phi_0 = \frac{\phi_H + \phi_L}{2}$. $\phi_H$ and $\phi_L$ represent the interface indicator values for the heavy and light fluids, respectively, set to 1.0 for the heavy fluid and 0.0 for the light fluid. The unit normal vector $\hat{n}$ for the interface can be defined as:

$$\hat{n} = \frac{\nabla \phi}{|\nabla \phi|}. \tag{3}$$

Note, the interface location at $x_0$ is initialized as Yan & Zu (2007):

$$\phi(x) = \phi_0 \pm \frac{\phi_H - \phi_L}{2} \tanh(\frac{|x - x_0|}{\xi/2}). \tag{4}$$

According to the phase-field model, the following equations exist for incompressible multiphase flows (Ding et al. (2007); Li et al. (2012)):

$$\frac{\partial \rho}{\partial t} + \nabla \cdot (\rho u) = 0, \tag{5a}$$

$$\rho \left( \frac{\partial u}{\partial t} + u \cdot \nabla u \right) = -\nabla p + \nabla \cdot \left( \mu \left[ \nabla u + (\nabla u)^T \right] \right) + F_s + F_b. \tag{5b}$$

In Equation 5a, $\rho$ represents the density of fluids, $p$ denotes the macroscopic pressure, $F_b$ is the body force, and $F_s$ corresponds to the surface tension force. The equation for calculating the surface tension force term is also expressed as Jamet et al. (2002):

$$F_s = \mu_\phi \nabla \phi, \tag{6}$$

where

$$\mu_\phi = 4\beta\phi(\phi - 1)(\phi - 1/2) - \kappa\nabla^2\phi, \tag{7}$$

denotes the chemical potential equation utilized for binary fluids (JACQMIN, 2000). Equation 8 establishes a relation between the coefficients $\beta$ and $\kappa$, interface thickness $\xi$, and surface tension $\sigma$, as;

$$\beta = 12\sigma/\xi, \quad \kappa = 3\sigma\xi/2. \tag{8}$$

## A.2 LATTICE BOLTZMANN METHOD

Given that interfaces are typically of mesoscopic scale, the kinetic-based Lattice Boltzmann Method (LBM) presents a more effective approach for simulating multiphase flows compared to the traditional Navier-Stokes solvers (Sukop & Thorne, 2006; Huang et al., 2015). The Chapman-Enskog analysis validates the consistency between the LBM and the Navier-Stokes equations (Krüger et al., 2017). In this study, we investigate hydrodynamic properties such as velocity and pressure using the standard form of the Lattice Boltzmann equation as outlined in Guo et al. (2002):

$$f_a\left(x + e_a\delta t,\ t + \delta t\right) = f_a\left(x, t\right) + \Omega_a(x, t) + F_a(x, t), \tag{9}$$

In this context, $f_a$ denotes the velocity-based hydrodynamic distribution function for incompressible fluids, $\Omega_a$ represents the collision operator, and $F_a$ signifies the force term. This study employs the two-dimensional nine-velocity (D2Q9) model for 2D simulations and the three-dimensional nineteen-velocity (D3Q19) model for 3D simulations.

To define the interface between phases, we employed the following Lattice Boltzmann Equation (LBE) to accurately determine the interface between fluid phases (Geier et al., 2015):

$$g_a\left(x + e_a\delta t, t + \delta t\right) = g_a\left(x, t\right) - \frac{g_a\left(x, t\right) - \bar{g}_a^{eq}\left(x, t\right)}{\tau_\phi + 1/2} + F_a^\phi(x, t). \tag{10}$$

Here, $g_a$ represents the distribution function for the phase-field, and $\tau_\phi$ denotes the dimensionless phase-field relaxation time. The forcing term is calculated as follows:

$$F_a^\phi\left(x, t\right) = \delta t \frac{\left[1 - 4(\phi - \phi_0)^2\right]}{\xi}\omega_a e_a \ \cdot \ \frac{\nabla\phi}{|\nabla\phi|}. \tag{11}$$

In Equation 11, $\omega_a$ and $e_a$ denote the weight coefficient and the mesoscopic velocity set, respectively. Here, $\xi$ denotes the thickness of the interface. As illustrated in Figure 8, we carefully selected this parameter to ensure adequate lattice nodes within the interface. This choice is critical for accurately capturing the complex physics in the rapid change of material properties across the interface. The appropriate selection of $\xi$ ensures that the computational mesh can effectively represent the gradients and variations within the interface, thus enhancing the overall stability and accuracy of the simulation.

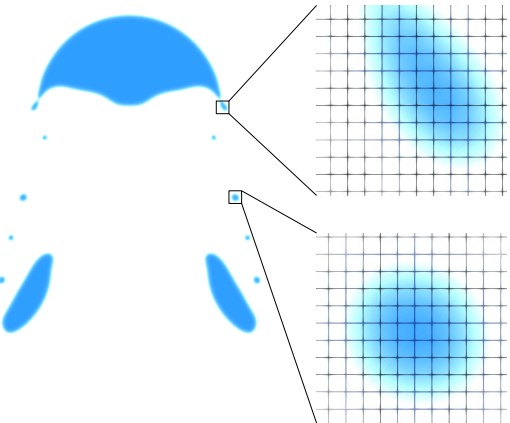

**Figure 8:** *Illustration of the interface region captured by the computational mesh. The magnified views show the distribution of lattice nodes within the interface, ensuring precise resolution of interfacial dynamics and transitions. The careful selection of the interface thickness parameter $\xi$ ensures that the mesh adequately represents the gradients and variations in the interface region.*

## A.3 VALIDATION

In this section, we validate our numerical model through benchmark tests covering a range of two-phase flow phenomena. We include four distinct validation cases to comprehensively assess the

accuracy and robustness of our approach: (1) the capillary wave problem, which evaluates the model's capability to handle surface tension-driven flows; (2) the bubble rising dynamics, which tests the interaction between buoyancy and viscous forces; (3) the falling droplet dynamics, which examines the breakup mechanisms of liquid droplets under gravity; and (4) the Rayleigh-Taylor instability, which explores the interfacial instability between fluids of differing densities under gravitational influence. Each subsection compares our simulation results and established experimental or numerical data, demonstrating the model's fidelity across various flow regimes.

### A.3.1 CAPILLARY WAVE

To validate our Lattice Boltzmann Method (LBM) simulations of two-phase flow, we focus on the dynamic behavior of capillary waves at the interface between two immiscible fluids. In our study, a sinusoidal perturbation with a small amplitude $\eta_0$ and wave number $k$ is applied to the initially quiescent interface. This setup provides a rigorous test for the LBM framework, as it has a well-established analytical solution for cases with identical kinematic viscosities $\nu$ but differing densities of the two fluids. The temporal evolution of the interface amplitude $\eta(t)$ is utilized as a benchmark for our simulations. The analytical expression for the decay of the wave amplitude, $\eta(t)$, is given by Prosperetti (1981):

$$\frac{\eta(t)}{\eta_0} = \frac{4(1-4\gamma)\nu^2 k^4}{8(1-4\gamma)\nu^2 k^4 + \omega_0} \operatorname{erfc}(\sqrt{\nu k^2 t}) + \sum_{i=1}^{4} \frac{z_i}{Z_i} \frac{\omega_0^2}{z_i^2 - \nu k^2} e^{(z_i^2 - \nu k^2)t} \operatorname{erfc}(z_i \sqrt{\nu t}) \quad (12)$$

where $\omega_0 = \sqrt{\frac{\sigma k^3}{\rho_H + \rho_L}}$ is the angular frequency, $\gamma = \frac{\rho_H \rho_L}{(\rho_H + \rho_L)^2}$ and $Z_i = \prod_{\substack{1 \le j \le 4 \\ j \ne i}} (z_j - z_i)$. The evaluation of the complementary error function $\operatorname{erfc}(z_i)$ can be done by solving the following algebraic equation:

$$z^4 - 4\gamma\sqrt{\nu k^2} z^3 + 2(1-6\gamma)\nu k^2 z^2 + 4(1-3\gamma)(\nu k^2)^{3/2} z + (1-4\gamma)\nu k^2 + \omega_0^2 = 0. \quad (13)$$

Our validation involves analyzing the propagation of capillary waves, an inherently transient process that tests the model's ability to accurately capture key physical parameters such as density and viscosity ratios, along with surface tension effects. By varying these parameters and the wavelength, we compare the simulation results with predictions from linear theory. According to Figure 9, the lighter fluid with density $\rho_L$ overlays the heavier fluid with density $\rho_H$, with the initial interface described by $y = L + \eta_0 \cos(2\pi x)$, where $\eta_0$ is the initial perturbation amplitude. The decay of this wavy profile to a flat interface, driven by viscosity and surface tension, without external forces like gravity, serves as a critical validation test for our LBM approach. The computational domain is

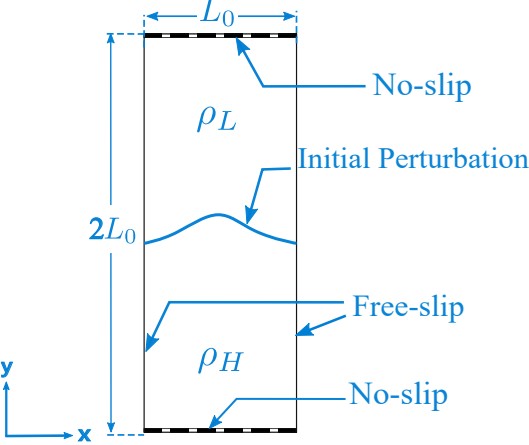

**Figure 9:** *Schematic diagram of the capillary wave problem setup.*

discretized into a grid of 256 by 512 lattice nodes. Free-slip boundary conditions are applied in the direction of wave propagation, while no-slip conditions are imposed at the top and bottom boundaries. The simulation parameters are set as follows: $\eta_0 = 0.02$, $\sigma = 10^{-4}$, $\xi = 4$, and $M_\phi = 0.02$. Since the interface may not align exactly with the grid points, the values of $\eta(t)$ are interpolated from $\phi$ values using the following relationship:

$$\eta(t) = y - \frac{\phi(x_{L0/2}, y)}{\phi(x_{L0/2}, y) - \phi(x_{L0/2}, y - 1)}, \quad \phi(x_{L0/2}, y)\phi(x_{L0/2}, y - 1) < 0. \quad (14)$$

The length ($\eta$) and time scales ($t$) are normalized by the initial amplitude $a_0$ and the angular frequency $\omega_0$, respectively, denoted as $\eta^* = \eta/\eta_0$ and $t^* = t\omega_0$.

It is worth noting that angular frequency is crucial for any wave system. It depends on surface tension, viscosity, wave number, and density values. The equation is derived assuming that both fluids have the same viscosity, set to $\nu = 0.005, 0.0005$. Note that the wavelength magnitude matches the grid size $L_0 = 256$.

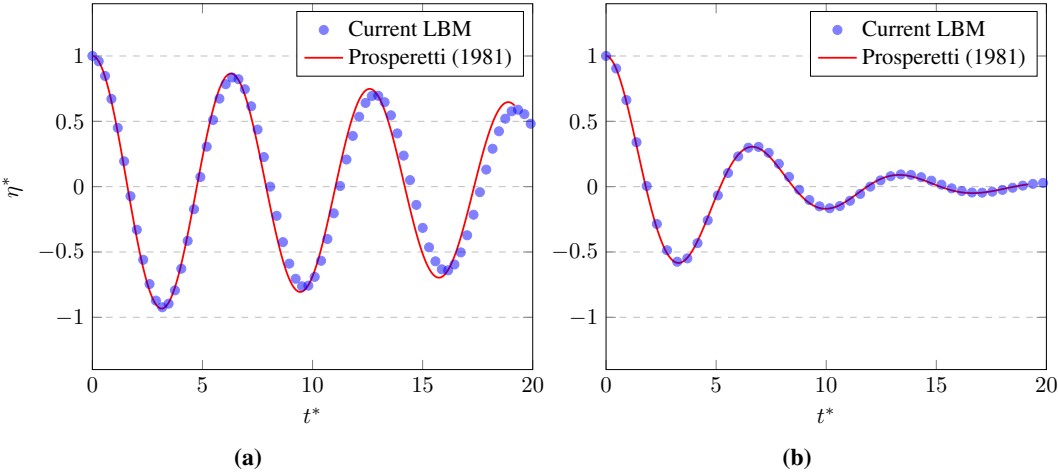

**Figure 10:** *Comparison of the normalized interface amplitude $\eta^*$ as a function of normalized time $t^*$ between the current LBM simulation and the analytical solution by Prosperetti (1981). (a) corresponds to a viscosity of $\nu = 0.0005$, and (b) corresponds to a viscosity of $\nu = 0.005$. The LBM results (blue circles) closely match the analytical results (red line).*

### A.3.2    RISE OF A SINGLE BUBBLE IN QUIESCENT FLUID

The dynamics of a rising bubble have been extensively studied due to their significance in various natural and industrial processes. When a bubble rises through a liquid, it is subjected to several forces, including buoyancy, drag, and surface tension, which influences its shape, velocity, and trajectory (Bhaga & Weber, 1981b; Amaya-Bower & Lee, 2010; Hua & Lou, 2007; Khanwale et al., 2023). Our investigation focuses on the dynamics of a bubble rising within a rectangular channel. The simulation begins with a circular bubble of diameter $D = L_0/5$ placed at the coordinates $(L_0/2, L_0/2)$ within a domain with a length of $L_0$ and a height of $4L_0$. Boundary conditions are set such that the no-slip is applied at the top and bottom, while free-slip boundary conditions are used for the lateral boundaries. The fluids experience a volumetric buoyancy force $F_b = -(\rho - \rho_h)g_y\mathbf{j}$, where $g_y$ represents the gravitational acceleration in the $y$-direction. This study highlights four crucial dimensionless parameters: the density ratio $\rho_h/\rho_l$, the viscosity ratio $\mu_h/\mu_l$, the gravity Reynolds number, and the Eötvös (Bond) number.

The gravity Reynolds number is defined as:

$$\mathrm{Re}_h = \frac{\sqrt{g_y\rho_h(\rho_h - \rho_l)D^3}}{\mu_h} \quad (15)$$

The Eötvös (Bond) number is defined as:

$$\text{Eo} = \frac{g_y(\rho_h - \rho_l)D^2}{\sigma} \tag{16}$$

In many studies, the Morton number is also considered, defined as:

$$\text{Mo} = \frac{g_y(\rho_h - \rho_l)\mu_h^4}{\sigma^3 \rho_h^2} \tag{17}$$

The dimensionless time is also defined by:

$$t^* = t\sqrt{\frac{g_y}{D}} \tag{18}$$

The reference velocity scale needed in the Péclet number can be chosen for gravity-driven flows as $U_0 = \sqrt{g_y D}$. Four sets of simulations are conducted at Four different Eötvös and Morton numbers. The density and viscosity ratios are fixed at 1000 and 100, respectively. The numerical parameters are $L_0 = 512$, $\text{Pe} = 25$ and $\text{Cn} = 0.010$, and the LBM simulation results are shown in Figure 11.

To evaluate the accuracy and reliability of the proposed LBM, a comparison is made between the results obtained from the LBM approach and those from the experiments and FVM, as illustrated in Figure 11. In the spherical regime, surface tension dominates, resulting in small bubbles that maintain a nearly spherical shape due to the strong cohesive forces at the interface. As the bubble size increases, the shape transitions to an ellipsoidal form. In this ellipsoidal regime, the inertial forces become more significant, causing the bubble to deform. This deformation is influenced by the surrounding liquid's viscosity and the interface's surface tension. The dynamics of this regime can be described using correlations that account for the balance between inertial and surface tension forces (Amaya-Bower & Lee, 2010). In the spherical cap regime, the bubbles are large enough that inertia forces dominate, leading to further deformation into a cap shape. This regime is characterized by a significant increase in terminal velocity, which is proportional to the size of the bubble (Bhaga & Weber, 1981a). These patterns are consistent among all results.

### A.3.3 FALLING DROPLET

The dynamics of a falling droplet under gravity is another fascinating two-phase flow phenomenon that has been extensively studied in the literature (Yang et al., 2021; Jalaal & Mehravaran, 2012). In this study, a liquid droplet with diameter $D = L_0/5$ is initially placed at $(L_0/2, 6L_0/2)$ within a rectangular computational domain of length $L_0$ and height $3L_0$. The same boundary conditions are applied as in the bubble rising simulations: the no-slip boundary condition is applied at the top and bottom, while free-slip boundary conditions are imposed at the lateral boundaries. The volumetric buoyancy force $F_b = -(\rho - \rho_l)g_y\mathbf{j}$, where $\mathbf{j}$ is unit vector in $y$-direction and $g_y$ represents the gravitational acceleration in the $y$-direction, acts on the fluids.

The dimensionless analysis identifies several key parameters that characterize the flow: the density ratio $\rho_h/\rho_l$, the viscosity ratio $\mu_h/\mu_l$, the gravity Reynolds number, and the Eötvös (Bond) number. The gravity Reynolds number is defined as:

$$\text{Re}_h = \frac{\sqrt{g_y\rho_h(\rho_h - \rho_l)D^3}}{\mu_h} \tag{19}$$

Similarly, the Eötvös number, which represents the ratio of gravitational forces to surface tension forces, is given by:

$$\text{Eo} = \frac{g_y(\rho_h - \rho_l)D^2}{\sigma} \tag{20}$$

Another important dimensionless group in the literature is the Morton number, which characterizes the fluid properties affecting the bubble and droplet dynamics:

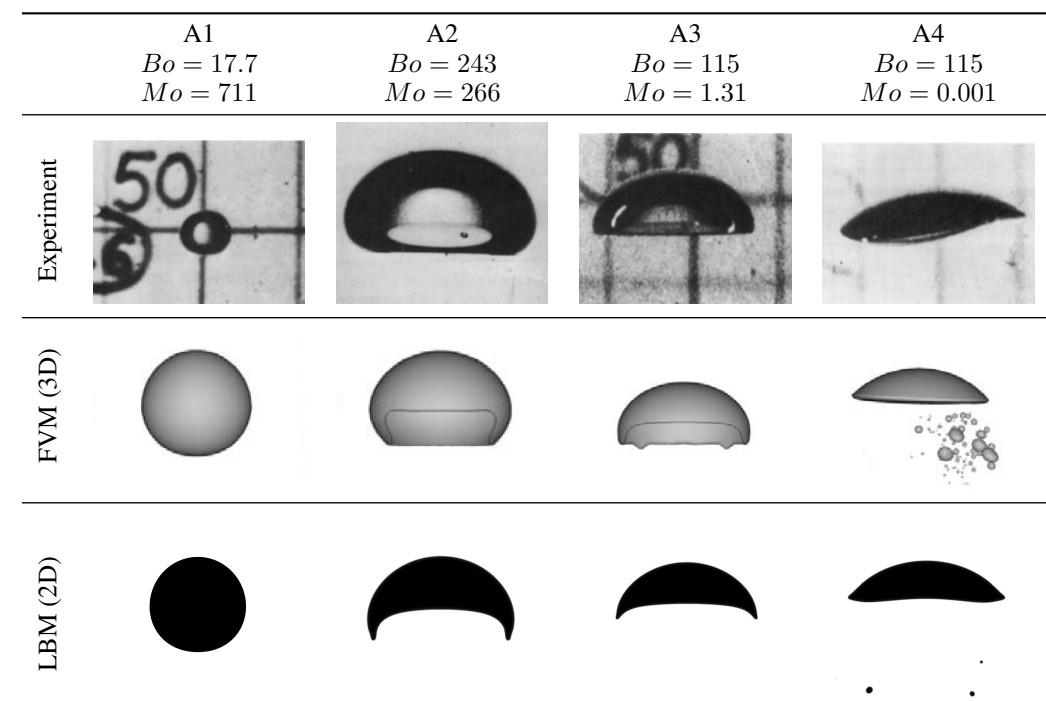

**Figure 11:** *Comparison of bubble shapes at constant rise velocity: Experimental results by Bhaga & Weber (1981a), LBM results, and FVM results by Gumulya et al. (2016) for various Bond numbers (Bo) and Morton numbers (Mo).*

$$\text{Mo} = \frac{g_y(\rho_h - \rho_l)\mu_h^4}{\sigma^3 \rho_h^2} \tag{21}$$

The Ohnesorge number (Oh) is a dimensionless number that characterizes the relative importance of viscous forces compared to inertial and surface tension forces in a fluid. It is particularly relevant in the study of droplet dynamics and is defined as:

$$\text{Oh} = \frac{\mu_h}{\sqrt{\rho_h \sigma D}} \tag{22}$$

The simulation is conducted at a moderate density ratio to capture the breakup mechanisms of the falling droplet, allowing for comparisons with the VOF model. The simulation considers an Eötvös number: $Eo = 288$, with density and viscosity ratios fixed at 10 and 1, respectively, and the Oh number set to 0.05. The numerical parameters are $Pe = 5$ and $Cn = 0.010$. As mentioned in Section A.3.2, the reference velocity scale needed for the Péclet number can be chosen as $U_0 = \sqrt{g_y D}$ for gravity-driven flows. Also, dimensionless time can be defined by:

$$t^* = t\sqrt{\frac{g_y}{D}} \tag{23}$$

Our simulation results exhibit excellent agreement with the findings of Jalaal & Mehravaran (2012). As shown in Figure 12, the comparison of the deformation of a liquid drop using both the Lattice Boltzmann Method (LBM) in 2D and the Volume of Fluid (VOF) method in 3D demonstrates that the evolution of the drop shapes over time is remarkably similar. For instance, at $t^* = 0.1647$, both methods capture the formation of a curved interface, and at $t^* = 0.3575$, the drop breakup into smaller droplets is observed in both approaches. This consistency across different numerical methods, with parameters set at $Eo = 288$, $Oh_h = Oh_l = 0.05$, and $\rho^* = 10$, validates the robustness and accuracy of our LBM simulations in replicating complex two-phase flow phenomena.

Overall, the dynamics of falling droplets involve complex interactions between buoyancy, inertia, and surface tension forces, leading to various deformation and breakup patterns, such as forming bags, ligaments, and secondary droplets. These phenomena are influenced significantly by the Eötvös number, with higher values leading to more pronounced deformations and faster breakup processes (Jalaal & Mehravaran, 2012).

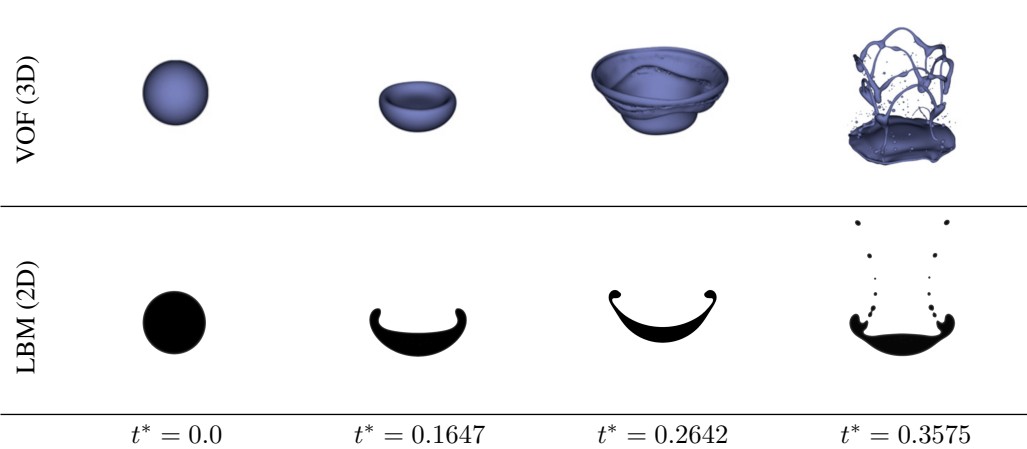

**Figure 12:** *Comparison of the deformation of a liquid drop using the LBM and VOF methods: Current results and those of Jalaal et al. (2012) (Jalaal & Mehravaran, 2012) for $Eo = 288$, $Oh_h = Oh_l = 0.05$, and $\rho^* = 10$.*

### A.3.4 RAYLEIGH-TAYLOR INSTABILITIES

The Rayleigh-Taylor instability (RTI) arises when a denser fluid is positioned above a less dense fluid in the presence of a gravitational field, causing the interface between the two fluids to become unstable. This phenomenon has been extensively studied due to its relevance in various natural and engineering contexts (Khanwale et al., 2023; Ren et al., 2016; Zu & He, 2013).

We consider a computational domain of size $[0, L_0] \times [0, 4L_0]$ with $L_0 = 256$ for our simulations. The initial interface is defined as $y_0(x) = 2L_0 + 0.1L_0 \cos(2\pi x/L_0)$. Periodic boundary conditions are applied on the left and right boundaries, while no-slip conditions are enforced at the top and bottom boundaries. The dimensionless numbers characterizing the RTI include the Atwood number, Reynolds number, Capillary number, and Peclet number:

$$At = \frac{\rho_H - \rho_L}{\rho_H + \rho_L}, \tag{24}$$

$$Re = \frac{\rho_H U_0 L_0}{\mu_H}, \tag{25}$$

where $U_0 = \sqrt{g_y L_0}$,

$$Ca = \frac{\mu_H U_0}{\sigma}, \tag{26}$$

$$Pe = \frac{U_0 L_0}{M}. \tag{27}$$

In our study, we used a density ratio $\rho^* = 3$, viscosity ratio $\mu^* = 1$, Reynolds number $Re = 128$, Atwood number $At = 0.5$, Peclet number $Pe = 744$, and interface width $\xi = 5$. The results are compared with the findings from Ren et al. (2016) and Zu & He (2013). The dimensionless time is defined as $t^* = t/t_0$, where $t_0 = \sqrt{L_0/(gAt)}$.

Snapshots of the interface evolution for the 2D Rayleigh-Taylor instability at different times are shown in Figure 13. Initially, the interface undergoes a symmetrical penetration of the heavier fluid into the lighter fluid, forming counter-rotating vortices. As time progresses, the heavier fluid rolls up into mushroom-like shapes, and secondary vortices form at the tails of the roll-ups. Our simulations' interface patterns and vortex structures are consistent with those reported in previous studies (Zu & He, 2013; Ren et al., 2016).

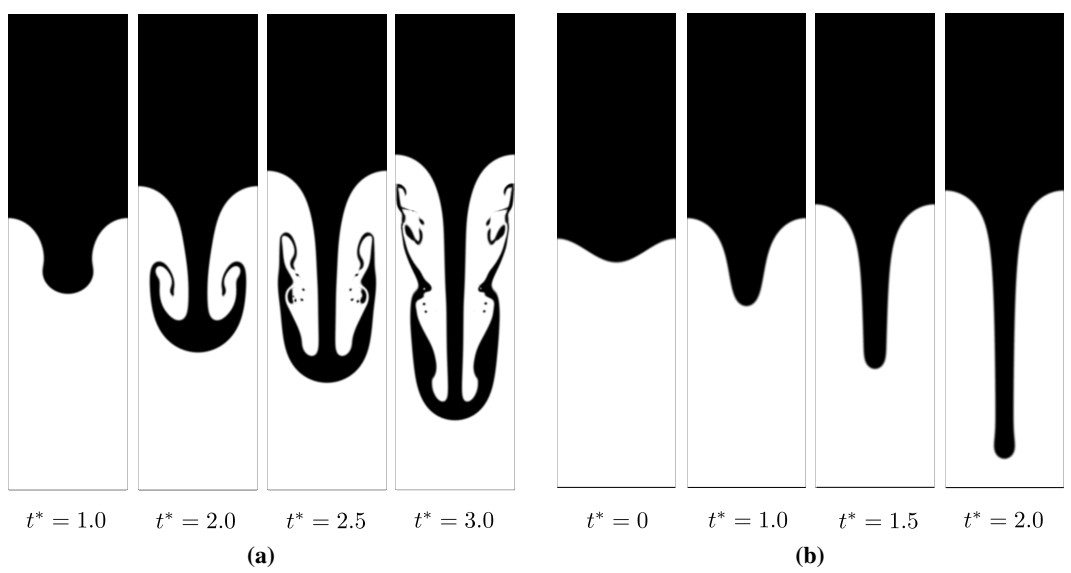

| $t^* = 1.0$ | $t^* = 2.0$ | $t^* = 2.5$ | $t^* = 3.0$ | $t^* = 0$ | $t^* = 1.0$ | $t^* = 1.5$ | $t^* = 2.0$ |
|---|---|---|---|---|---|---|---|

(a)                                        (b)

**Figure 13:** *Evolution of the interface pattern of the 2D Rayleigh-Taylor instability for two scenarios: (a) $\rho^* = 3$, $\mu^* = 1$, $Re = 128$, $At = 0.500$, $Pe = 744$, $\xi = 5$; (b) $\rho^* = 1000$, $\mu^* = 100$, $Re = 3000$, $At = 0.998$, $Pe = 200$, $Ca = 8.7$, $\xi = 5$.*

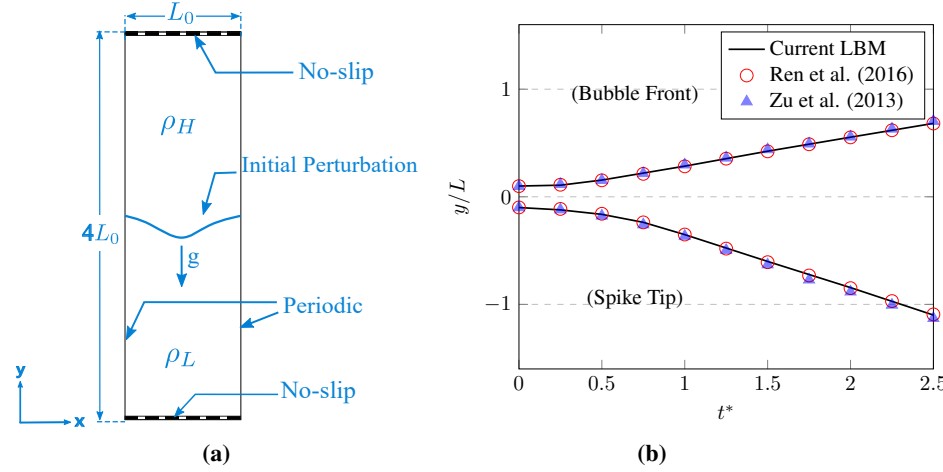

(a)                                        (b)

**Figure 14:** *(a) Schematic of the initial setup for the Rayleigh-Taylor instability simulation, showing the boundary conditions and initial perturbation. (b) Comparison of the bubble front and spike tip positions over time for the Rayleigh-Taylor instability case with parameters $\rho^* = 3$, $\mu^* = 1$, $Re = 128$, $At = 0.500$, $Pe = 744$, and $\xi = 5$. The current LBM results (solid line) are compared with the results of Ren et al. (2016) (red circles) and Zu & He (2013) (blue triangles), showing excellent agreement in capturing the evolution of the instability.*

