# OpenReview forum: "MPFBench: A Large Scale Dataset for SciML of Multi-Phase-Flows: Droplet and Bubble Dynamics"
_ICLR.cc/2025/Conference — Submitted to ICLR 2025_

### Official Review · Reviewer_PV15 · 2024-10-24

**Soundness:** 2
**Presentation:** 3
**Contribution:** 2
**Rating:** 3
**Confidence:** 3

**Summary:**

MPF-Bench provides a dataset that contains 11,000 2D and 3D simulations of challenging multiphase fluid dynamics, and is produced with a well-validated Lattice Boltzmann method framework. Six neural operators and foundation models are trained and evaluated on six different datasets, including both sequence-to-field and sequence-to-sequence tasks. The authors finally get encouraging results which show that machine learning approaches are able to learn multiphase fluid dynamics.

**Strengths:**

1. MPF-Bench provides a large number of snapshots.
2. The simulation framework is well-validated.
3. Six models are evaluated,

**Weaknesses:**

1. 'Our models are trained on a random selection of 1000 samples from the bubble dataset.': Does this mean that all models are trained with the **2D** bubble dataset? Are these 1000 samples fixed for different models?
2. It would be meaningful to explore the results of models on different types of datasets and compare them (e.g., bubble and droplet, different difficulty due to different parameters), which will help us figure out which model is better at handling which case.
3. The evaluation of models on the 3D datasets will be closer to the real world and more interesting.
4. More visualizations of different models will be more convincing and help better compare their performance.
5. l156: It is better to explain what each symbol in the equations refers to.
6. l168: Do not need ".
7. l307: [resolution_z] is incorporated...
8. l310: families.

**Questions:**

1. How do you train the foundation models (scOT and Poseidon)? Are they retrained or fine-tuned based on pre-trained checkpoints?

---

> ### Author Response · Authors · 2024-11-27
>
> ## Response:
> To address the first weakness noted, all models in this study are trained and tested on the same set of 1,000 samples from the 2D bubble rise dataset to ensure consistency in evaluation. As you suggested, in future work, we plan to expand our analysis to include three levels of difficulty within the 2D dataset (*easy, moderate, and hard*). This will allow us to better understand model performance across varying levels of complexity and demonstrate the ability of SciML models to learn and generalize transient two-phase flow phenomena.
>
> As you pointed out, incorporating 3D physics would provide valuable insights into real-world problems. However, the 3D dataset poses additional challenges, as it is computationally expensive and requires data-parallel training to accommodate sequences of timesteps. In future work, we plan to explore approaches such as data-parallel training to evaluate models under different scenarios for the complex phenomena occurring in 3D.
>
> Additionally, we are sharing this dataset with the SciML community to encourage further research and the development of models capable of addressing higher-dimensional, transient two-phase flow problems. We will ensure that all typos are corrected in the camera-ready version and will carefully address the points raised in this feedback.
>
>
>
> ## Questions:
> > 1- How do you train the foundation models (scOT and Poseidon)? Are they retrained or fine-tuned based on pre-trained checkpoints?
>
> scOT is trained from scratch, while Poseidon is fine-tuned based on a pre-trained checkpoint.

---

> > ### Comment · Reviewer_PV15 · 2024-11-28
> >
> > Thank you for the response. I believe adding these technical details and conducting broader experiments would greatly enhance the paper. However, at its current stage, the paper still feels incomplete, so I will maintain my score.

---

### Official Review · Reviewer_beKU · 2024-11-01

**Soundness:** 3
**Presentation:** 3
**Contribution:** 3
**Rating:** 8
**Confidence:** 3

**Summary:**

Good paper on a useful physics dataset. Multiphase flows represent a frontier domain in flow physics. Time-series datasets are also useful across different ML domain including video modeling, etc.

I only have questions that would help clarify some context and experimental choices for better presentation.

Edit 1: Concerns have been addressed. Raising score to 8 to recommend for strong acceptance.

**Strengths:**

1. Time-dependent multiphase data -- rich dataset.
2. Extensive model evaluation
3. Good lit review of previous work
4. Lattice boltzmann solvers are high fidelity
5. 4000 GPU hours is substantial
6. Good Qualititative demonstration of ML predictions
7. Solid Appendix
8. Good reproducibility efforts.

**Weaknesses:**

1. Applications of this dataset are not obvious -- could be emphasized more in introduction or via eval demonstrations
2. Description of physics methods requires a bit more clarity for non-physics readers in this general ML venue.
3. Connection to anonymous repo had 522 timeout when I clicked -- I assume that this will be fixed after double blind review.

**Questions:**

1. In appendix A, can you briefly describe the Allen Cahn equations a bit more for the readers? Specifically on big picture descriptions on how close is this to direct numerical simulation of Navier Stokes?
2. In appendix A, what's the benefit of Lattice Boltzman methods vs conventional interface-capturing Finite Volume Solvers? Are there any cost-accuracy tradeoffs with your simulation approach? This could be useful for readers to know as well.
3. Since Section 4, line 365. How were hyper parameters chosen?
4. Section 4 and 5 -- How many train/val/test splits?
5. Can you spend a paragraph or 2 explaining the broader applications of this dataset and importance of sequence to field and sequence predictions benchmarks in the intro?

---

> ### Author Response · Authors · 2024-11-27
>
> ## Response:
> Thank you for your feedback. We have revised the manuscript to better emphasize the dataset's real-world applications, such as optimizing oxygen transfer in wastewater treatment and improving oil spill cleanup strategies, highlighting its relevance for industries reliant on multiphase flow modeling. To improve clarity for non-physics readers, we have expanded the description of the physics methods in the appendix, including why the Lattice Boltzmann Method and Allen-Cahn equations were chosen. We have also simplified technical explanations to make them more accessible to a general ML audience. Lastly, the anonymous repository [link](https://huggingface.co/datasets/mshad2345/MPFBench) has been tested and is working properly. We hope these updates address your concerns.
>
> ## Questions:
>
> > 1- In appendix A, can you briefly describe the Allen Cahn equations a bit more for the readers? Specifically on big picture descriptions on how close is this to direct numerical simulation of Navier Stokes?
>
> ### Allen-Cahn Equation and DNS of Navier-Stokes
>
> The **Allen-Cahn equation** models phase separation and interface dynamics using an order parameter $(\phi)$. It simplifies interface tracking by representing interfaces as diffuse regions, avoiding explicit reconstruction. While it is not a standalone DNS of Navier-Stokes, it can be coupled with Navier-Stokes to simulate two-phase flows.
>
> - **Strengths**: Efficient interface handling, natural inclusion of surface tension, and robust merging/splitting dynamics.
> - **Limitations**: Artificial interface thickness $( \epsilon )$ can smooth out small-scale phenomena, limiting accuracy in high-Reynolds-number flows.
>
> It is an efficient approximation for complex two-phase flows but less detailed than pure DNS.
>
>
>
>
>
> > 2- In appendix A, what's the benefit of Lattice Boltzman methods vs conventional interface-capturing Finite Volume Solvers? Are there any cost-accuracy tradeoffs with your simulation approach? This could be useful for readers to know as well.
>
> **Lattice Boltzmann Method (LBM)** offers simplicity, efficient interface tracking, and scalability due to its local operations. Unlike **Finite Volume Solvers (FVS)**, LBM avoids complex equation coupling and explicit interface tracking.
>
> - **Benefits**: Easy parallelization, natural multiphysics integration, and reduced complexity.
> - **Tradeoffs**: LBM is less accurate for sharp interface or high-Reynolds flows, while FVS achieves higher accuracy but is costlier for complex flows.
>
> LBM is ideal for scalable, efficient multiphase simulations, with some tradeoffs in interface resolution and accuracy.
>
>
> > 3- Since Section 4, line 365. How were hyper parameters chosen?
>
> The hyperparameters were carefully chosen through an extensive search on the dataset. We explored a wide range of values for parameters such as learning rate, number of layers, and hidden dimensions to identify the optimal configuration for our benchmarks. Details of the selected hyperparameters will be provided in the appendix for better reproducibility of our results.
>
> > 4- Section 4 and 5 -- How many train/val/test splits?
>
> The dataset was split into 70% for training, 10% for validation, and 20% for testing. We conducted experiments using a single train/val/test split.
>
> > 5- Can you spend a paragraph or 2 explaining the broader applications of this dataset and importance of sequence to field and sequence predictions benchmarks in the intro?
>
> We have revised the introduction to emphasize the broader applications of this dataset and expanded on the significance of sequence-to-field and sequence-to-sequence prediction benchmarks. Sequence-to-field predictions are essential for modeling steady-state behaviors, where the goal is to map an input sequence (e.g., parameters describing initial or boundary conditions) directly to the corresponding field outputs. In contrast, sequence-to-sequence predictions are designed to capture the transient dynamics of bubble rise by mapping an input sequence (e.g., time-series data describing earlier states of the system) to another sequence representing the system’s future states. This technique leverages concepts rooted in natural language processing, where seq2seq models transform one sequence into another through an encoder-decoder architecture. Originally developed for tasks like machine translation, seq2seq methods are well-suited for time-evolving physical systems as they can learn complex temporal dependencies. For example, in our dataset, seq2seq predictions enable accurate modeling of time-varying behaviors such as bubble deformation, oscillations, and breakup, which are critical for understanding and simulating multiphase flow phenomena.

---

> > ### Comment · Reviewer_beKU · 2024-12-03
> >
> > Thank you for addressing these concerns. Raising score to 8 for strong acceptance

---

### Official Review · Reviewer_iniK · 2024-11-04

**Soundness:** 4
**Presentation:** 3
**Contribution:** 3
**Rating:** 3
**Confidence:** 4

**Summary:**

MPF-Bench provides a large dataset of two multiphase fluid flow simulation types, namely rising bubble and falling droplet dynamics. The paper includes the performance of six popular neural network baselines, covering both sequence-to-field and sequence-to-sequence predictions.

**Strengths:**

1) Dataset accuracy is ensured by providing validation studies for the LBM solver used to generate the datasets.
2) The work is overall well presented and the limitations are discussed.

**Weaknesses:**

On a high level, I see how multi-phase flow simulations can be useful in industry, but I'm not convinced that the proposed dataset is practically useful. Typically, industrial applications involve at least one of (a) interactions between bubbles, (b) thermal exchange with the environment, (c) interactions with walls, or (d) complex geometries. I would suggest adding such more practically relevant cases, i.e., if a task of interest is cavitation at turbine blades, one could simulate bubbles next to a metal surface. The currently proposed dataset is relatively simple in its setup, and I would describe it as a scaled-up version of BubbleML.

Ideas for improvement of the manuscript:
1) Consider rewriting line 85 as there has been a predecessor for multiphase benchmarking using neural operators, i.e., BubbleML. For example, you could rewrite the sentence to "To our knowledge, only one study (Hassan et al., 2023) has evaluated the performance of neural operators on multiphase flows, and we are the first to evaluate a foundation model that has been pre-trained on single-phase data."
1) In line 137, mention how many datasets are for 2D and 3D separately.
1) In Section 3.3, line 261, the input field part could be rewritten. Does the input to the models include only the scalars?
1) In lines 269-270: Provide clarification regarding the timestep used to save the dataset. Is the timestep interval used to generate the dataset the same as the timestep of the LBM solver? Clarification on choosing this timestep coarsening factor would be insightful.
1) Provide the mathematical formula for the metrics (MSE and relative L2 error) in the appendix. Also, mention a reason for choosing these metrics and potentially discuss alternative, more physical metrics. By physical metrics, I mean something aligned with the downstream application, e.g., the error of the volume fraction of the bubble/droplet, the error in the velocity of the center of mass of the bubble/droplet, or any other relevant derived metric.
1) As a benchmarking paper, please consider providing standard deviations (over multiple seeds) of the metrics in Tables 5 and 6.
1) Providing details on the number of model parameters and hyper-parameters for each model used for benchmarking would be insightful.

**Questions:**

1) In line 268, by interface indicator, do the authors refer to the signed-distance function?
1) Looking at the figures in the manuscript, most flow fields seem symmetric with respect to the centered vertical axis, meaning that one could store only one half of the field in 2D and one quarter in 3D to describe the full flow field. Did you consider doing this to reduce the dataset size?

---

> ### Author Response · Authors · 2024-11-27
>
> ## Response:
> While the dataset may appear relatively simple in its setup, it has significant real-world applicability in industries where multiphase flow phenomena are critical. By capturing bubble rise dynamics across varying surface tensions, density ratios, and bubble sizes, it provides valuable insights into processes such as aeration in wastewater treatment, where bubble behavior directly influences oxygen transfer efficiency, and in chemical reactors, where rising bubbles play a vital role in enhancing mass and heat transfer. Additionally, the dataset has practical applications in environmental remediation, such as predicting bubble behavior in oil spill cleanup, aiding in the design of effective mitigation strategies. Although it does not currently include interactions between bubbles, thermal exchanges, or interactions with walls, the high-fidelity data it provides is a foundational resource for developing predictive models and optimizing processes in these and other multiphase flow applications. This focus on fundamental bubble dynamics allows researchers to validate and generalize their models for a wide range of industrial applications.
>
>
> The input to the models consists of the solution sequence for the input fields (velocity, pressure, and interface indicator) and does not include scalar quantities such as density ratio, viscosity ratio, Bond number, or Reynolds number. This distinction will be clarified in the text. The LBM solver was run for 400,000 timesteps, saving solutions every 4,000 timesteps to produce 100 uniformly distributed time snapshots. Details of the model hyperparameters and the mathematical formulas for the evaluation metrics (MSE and $L_2$ relative error) have been added to the appendix.
>
>
>
> ## Questions:
>
> > 1- In line 268, by interface indicator, do the authors refer to the signed-distance function?
>
> No, the interface indicator does not refer to the signed-distance function. Instead, it refers to the **volume fraction**, which serves as an indicator of the phase distribution within the domain. Specifically:
> - \( c = 0.5 \) indicates phase 1 (e.g., liquid),
> - \( c = -0.5 \) indicates phase 2 (e.g., gas),
> - \( c = 0 \) represents the interface between the two phases.
>
> This volume fraction approach is commonly used in multiphase flow simulations to distinguish between phases and to capture the interface without explicitly defining the geometry. It provides a smooth transition across the interface, enabling numerical methods to handle phase boundaries effectively.
>
>
> > 2- Looking at the figures in the manuscript, most flow fields seem symmetric with respect to the centered vertical axis, meaning that one could store only one half of the field in 2D and one quarter in 3D to describe the full flow field. Did you consider doing this to reduce the dataset size?
>
>
> While storing only half or a quarter of the fields could reduce dataset size, it introduces significant drawbacks. It adds complexity to data handling, requiring additional preprocessing and postprocessing to reconstruct full fields during training and inference, which complicates machine learning workflows. Furthermore, modern data compression techniques, such as the .npz format, already minimize dataset size effectively without compromising completeness or fidelity. Given these considerations, we opted to retain the full flow fields to ensure the dataset remains straightforward, accurate, and versatile for a wide range of applications.

---

> > ### Comment · Reviewer_iniK · 2024-12-03
> >
> > Similar to reviewer PV15, I still believe that there are missing pieces to this submission. Among others, **the parts from my weaknesses section which you did not address in your response**.
> >
> > Parts that you do mention in your response but I still disagree with include:
> > - Usefulness of the proposed datasets - ML is not good in out-of-distribution generalization, e.g., one cannot train on single bubble data and hope it generalizes to bubble interactions. I don't see how these datasets are useful for ML.
> > - Having symmetric flow fields and still storing all of it is just ridiculous.

---

### Official Review · Reviewer_QXki · 2024-11-04

**Soundness:** 3
**Presentation:** 3
**Contribution:** 3
**Rating:** 6
**Confidence:** 3

**Summary:**

This paper contributes a new large-scale dataset for droplet and bubble formation, which is important within the engineering and chemical process industry. The main contributions is a dataset larger than before and one which has been evaluated on state of the art SciML methods e.g FNO, UNet etc.

**Strengths:**

The main strength is the creation of a larger than previously available dataset and one which has been tested on numerous different state of the art ML methods. This is a very useful contribution to the AI4Science field that is strongly lacking in openly available datasets.

**Weaknesses:**

Whilst the paper (and the associated website) are well written, there are some aspects that are missing, particuarly in the paper. There is no discussion on the license of the dataset (a very important topic) within the paper itself. Looking at the sample data I found CC-BY-NC - which means no commerical usage. I would like to see a discussion on this in the main paper. Additionally there is limited discussion on how to actually use the data. On the website (hugging face) there are some steps on how to unzip and then a script to read in the data, but this should be in the appendix of the paper too.

**Questions:**

1) What is the license of the dataset (please justify your choice)
2) How do you plan to host the data (is it hugging face?)
3) What are you plans to ensure it's maintances over years and potentially decades.

---

> ### Author Response · Authors · 2024-11-27
>
> ## Response:
> We have addressed the feedback by explicitly including a discussion of the dataset's license (CC-BY-NC) in the main paper, emphasizing its purpose for non-commercial research and educational use within the SciML community to benchmark and challenge models on this complex dataset. Additionally, detailed instructions on how to download, unzip, and access the dataset have been added to the appendix, ensuring that the steps provided on the Hugging Face website are now fully documented within the paper for ease of use.
>
>
>
> ## Questions:
>
> > 1- What is the license of the dataset (please justify your choice)
>
> This dataset is intended for the SciML community to challenge state-of-the-art neural operators and foundation models and is licensed under **CC-BY-NC** for non-commercial use. The choice of this license ensures that the dataset is freely accessible for research and educational purposes while restricting its use for commercial applications without explicit permission, aligning with the open-access goals of the SciML community.
>
>
> > 2- How do you plan to host the data (is it hugging face?)
>
> The dataset will be hosted on **Hugging Face**, a widely used platform for machine learning datasets and models. Hugging Face provides:
> - Easy accessibility for researchers and practitioners.
> - A robust framework for organizing and managing large datasets.
> - Built-in version control to track updates and changes.
> - Seamless integration with popular machine learning tools and frameworks.
>
>
> > 3- What are you plans to ensure it's maintances over years and potentially decades.
>
> To ensure the dataset remains useful and relevant for years and decades, we have the following plans:
> 1. **Long-term Hosting on Hugging Face**:
>    - Hugging Face provides stable hosting and long-term accessibility to datasets.
>    - Regular updates to the repository will keep the dataset compatible with evolving machine learning frameworks.
>
> 2. **Community Involvement**:
>    - Engage the SciML community to provide feedback and contribute to dataset maintenance and improvements.
>    - Promote collaborative enhancements through GitHub tools linked to the dataset.
>
> 3. **Dataset Expansion**:
>    - Commit to periodic updates, including new 3D simulations and potentially three-phase flows (e.g., bubble rise and droplet fall in stratified flow).
>    - Expand the dataset to include more complex scenarios and physics to meet future research demands.
>
> 4. **Backup and Redundancy**:
>    - Maintain multiple backups of the dataset in secure storage locations to prevent data loss.

---

### Author Response · Authors · 2024-11-27
**General Response**

We thank the reviewers for their careful reviews and constructive suggestions. We are glad all reviewers found our paper well-written and recognized its contribution to addressing a key gap in the scientific machine learning (SciML) literature on multiphase flow modeling. Below, we summarize the updates made in response to the reviewers’ feedback.

We have explicitly discussed the dataset’s CC-BY-NC license in the main manuscript, emphasizing its intended use for non-commercial research and educational purposes. To improve accessibility, we have included detailed instructions for downloading, unzipping, and accessing the dataset in the appendix, complementing the guidance on the Hugging Face repository. A subset of the dataset can be accessed here: https://huggingface.co/datasets/mshad2345/MPFBench. Additionally, we have expanded the discussion of the dataset’s real-world applications to highlight its relevance in industries such as wastewater treatment and chemical reactors. The dataset’s high-fidelity simulations of bubble rise dynamics under varying surface tensions, density ratios, and bubble sizes provide foundational insights for optimizing oxygen transfer and predicting bubble behavior in oil spill cleanup.

Lastly, we have clarified the details of the machine learning training on 1000 samples used in the current study. Looking ahead, we plan to expand our analysis to include data from varying difficulty levels (easy, moderate, and hard) to gain a deeper understanding of SciML model generalization capabilities.

---

### Meta-Review · Area_Chair_Uvze · 2024-12-19

**Metareview:**

This submissions introduces a new dataset for CFD with the particularity of focusing on droplets and bubble formation. Generally speaking, creating publically available datasets is very useful for the community, and I'd like to thank the authors for their efforts. The reviewers also appreciated this, as well as the presentation of the paper and the presentation of its limitations.

However, several strong weaknesses were raised, in particular the simplicity of the studied physical processes; missing generalization to unseen situations; redundancy in data storage; missing information on the usage of the data, missing license (this was answered later).

While the authors attempted to answer some of the questions, they were quite handwavy on others and ignore some of the questions all together.

The AC sides with the critical reviewers and judges that the paper is not yet ready for publication.

**Additional Comments On Reviewer Discussion:**

The reviewers engaged with the authors

---

### Decision · Program_Chairs · 2025-01-22

Reject